

# A new species and new generic synonymy in the family Vietnamellidae (Insecta: Ephemeroptera) from mid-Cretaceous Burmese amber with notes on ancient dispersal across East Gondwana

Roman J. Godunko[1,2,3], Nadhira Benhadji[4], Alexander Martynov[5], Zhi-Teng Chen[6], Xuhongyi Zheng[7] and Arnold H. Staniczek[8]

[1] Biology Centre of the Czech Academy of Sciences, Institute of Entomology, České Budějovice, Czech Republic
[2] Department of Invertebrate Zoology and Hydrobiology, Faculty of Biology and Environmental Protection, University of Lodz, Łódź, Poland
[3] State Museum of Natural History, National Academy of Sciences of Ukraine, Lviv, Ukraine
[4] Institute of Technology and Life Sciences—National Research Institute, Raszyn, Poland
[5] National Museum of Natural History, National Academy of Sciences of Ukraine, Kyiv, Ukraine
[6] School of Grain Science and Technology, Jiangsu University of Science and Technology, Zhenjiang, China
[7] The Key Laboratory of Jiangsu Biodiversity and Biotechnology, College of Life Sciences, Nanjing Normal University, Nanjing, China
[8] Department of Entomology, State Museum of Natural History Stuttgart, Stuttgart, Germany

Corresponding author
Roman J. Godunko,
roman.hodunko@biol.uni.lodz.pl

## ABSTRACT

The monophyletic mayfly family Vietnamellidae has been introduced for a few extant species from the Indomalayan Realm. All these species belong to the genus *Vietnamella* and have been described in adult and larval stages. Recently, the fossil genus *Burmella* was established for male and female imagines of two new species from mid-Cretaceous Burmese amber and assigned to the family Vietnamellidae. In this contribution, we describe another species in the genus *Burmella*, namely *B. inconspicua* **sp. nov.** based on a female imago. It clearly differs from the previously described female of *B. clypeata* by the absence of an extension of the clypeus and by a different venation in fore and hind wings. Furthermore, we discuss here in detail the systematic position of the monotypic species *Burmaheptagenia zhouchangfai* originally assigned to the family Heptageniidae and provide arguments for the synonymy of the genus *Burmaheptagenia* syn. nov. with *Burmella*. We propose the new combination *Burmella zhouchangfai* **comb. nov.**, together with a modified generic diagnosis for the genus *Burmella* and discuss the adult characters of Vietnamellidae. Systematic placement of the genus *Burmella* in Vietnamellidae and the phylogenetic position of this family within Ephemerelloidea are discussed. We also consider the geographic origin of the family and the role of the Cretaceous Burma Terrane migration as a key event that facilitated the transfer of Gondwanan fauna to Asia.

## INTRODUCTION

*Vietnamella* Tshernova, 1972 was established as monotypic genus for the species *V. thani* Tshernova, 1972 described from larvae found in Vietnam. Besides this species, in the same publication, *Tshernova (1972)* described *Ephemerellina ornata* Tshernova, 1972 from China, which was later placed in the genus *Vietnamella* by *Wang & McCafferty (1995)*. *You & Su (1987)* and *Zhou & Su (1995)* described three other species from China, namely *Vietnamella dabieshanensis* You & Su, 1987, *V. guadunensis* Zhou & Su, 1995 and *V. qingyuanensis* Zhou & Su, 1995. However, *Zhou (2013)* subsequently placed *V. guadunensis* and *V. qingyuanensis* as junior synonyms of *V. dabieshanensis.* Even later, *Hu et al. (2017)* synonymised *V. dabieshanensis* with *V. sinensis* Hsu, 1936 (*Hsu, 1936*). Finally, *V. chebalingensis* Tong, 2020 from Guangdong Province, China, was described based on larval material (*Luo et al., 2020*). Also, *Luo et al. (2020)* discussed differences between Chinese and Thai species based on morphological and molecular data. A single record of larval *Vietnamella* sp. from India was published by *Sinha et al. (2018)*. Using material from Arunachal Pradesh in north-eastern India, specimens were described and illustrated, but not formally described and named as new species, so their designated holotype and paratypes have to be regarded as invalid. Considerable attention has been paid in the last few years to the diversity of the genus *Vietnamella* in Thailand. *Auychinda, Sartori & Boonsoong (2020)* initially published a review of Thai *Vietnamella* with description of a new species *V. maculosa* Auychinda, Sartori & Boonsoong, 2020, as well as the first description of the adult stages of *V. thani* based on material reared from larvae. Adults and larvae of a second new species from Thailand, *V. nanensis* Auychinda, Sartori & Boonsoong, 2020, were described using both morphological and genetic characters (*Auychinda et al., 2020*).

The first fossil representatives of Vietnamellidae were recorded by *Godunko, Martynov & Staniczek (2021)*, who, based on male and female imagines, described *Burmella paucivenosa* Godunko, Martynov & Staniczek, 2021 and *B. clypeata* Godunko, Martynov & Staniczek, 2021 in the newly established genus *Burmella* Godunko, Martynov & Staniczek, 2021. Analysing the systematic position of *Burmella*, they provided arguments to support their placement within Vietnamellidae, also listing diagnostic generic differences from extant *Vietnamella*.

The fossil mayflies from Cretaceous Burmese amber (Myanmar) are currently represented with nine families, only four of which are extinct (*Ross, 2024*). The genus *Burmella* is so far the single representative of Ephemerelloidea in Burmese amber.

Based on a review of earlier described taxa and new material, we show in the present contribution that fossil Vietnamellidae are far more diverse in the mid-Cretaceous Burmese (Myanmar) amber than previously assumed. We review the holotype of *Burmaheptagenia zhouchangfai* Chen & Zheng, 2023, a male imago, evaluate its systematic position, and show that this species rather belongs to the genus *Burmella* within Vietnamellidae, but not to Heptageniidae Needham, 1901 as proposed in the original description by *Chen & Zheng (2023)*. Thus, a new generic synonymy is established, *Burmella = Burmaheptagenia* Chen & Zheng, 2023 **syn. nov.**, with the new combination *Burmella zhouchangfai*

(Chen & Zheng, 2023) **comb. nov.** We also describe a new species *B. inconspicua* **sp. nov.** based on a female imago and discuss its relationships with the other three representatives of *Burmella*. Reviewing all species of *Burmella*, we also amend the imaginal diagnostic characters of *Burmella*, comparing them with extant species of the genus *Vietnamella*. Finally, we discuss the paleobiogeography of Vietnamellidae and consider the migration of this basal family of Ephemerelloidea across the Tethys Ocean *via* the Burmese terrane during the Cretaceous as an example of a significant event that contributed to the transfer of Gondwanan elements to Asia.

## MATERIALS & METHODS

### Material

The well-preserved specimen of the holotype of *B. inconspicua* **sp. nov.** is embedded in mid-Cretaceous Burmese amber. The stone originates from a mine situated in Northern Myanmar, Kachin State, Tanai Township, Hukawng Valley, SW of Tanai City, which corresponds to the findings of all other material of fossil Vietnamellidae (*Godunko, Martynov & Staniczek, 2021*), as well as many other Mesozoic mayfly taxa from mid-Cretaceous Burmese amber (*Chen & Zheng, 2023*).

Based on UePb zircon dating, the Hukawng amber can be assigned to the Early Cretaceous, Upper Albian, with a maximum age of 98.79 ± 0.62 Ma (*Shi et al., 2012*), which is equivalent to the earliest Cenomanian (*Gradstein, Ogg & Smith, 2004*).

The female imaginal holotype of *B. inconspicua* **sp. nov.** belonged to the Patrick Müller collection (Käshofen, Germany) (former inventory number BUB–3325) and was kindly donated to the State Museum of Natural History Stuttgart (SMNS) and filed under the inventory number SMNS BU–385. The male imaginal holotype of *Burmella paucivenosa* (inventory number SMNS BU-179) and holotype of female imago of *B. clypeata* (inventory number SMNS BU-321) are also both housed at SMNS (*Godunko, Martynov & Staniczek, 2021*). The male imaginal holotype of *Burmaheptagenia zhouchangfai* is deposited at the Insect Collection of Jiangsu University of Science and Technology, China (inventory number CZT-EPH-MA5).

### Specimen processing and imaging

The fossil material and comparative extant material was studied under a Leica M205 C (Leica Corporation, Wetzlar, Germany) and an Olympus SZX7 (Olympus Corporation, Tokyo, Japan) stereomicroscope. We used a Leica Z16 APO Macroscope equipped with a Leica DFC450 Digital Camera, using Leica Application Suite v. 3.1.8. to obtain stacked series of photographs with different levels of focus. Photo stacks were processed with Helicon Focus Pro 8.3.0 to obtain combined photographs with extended depth of field. Photographs were sharpened and adjusted in contrast and tonality in Adobe Photoshop™ (Adobe Systems Incorporated, San Jose, CA, USA). Drawings were made using a camera lucida on a Leica M205 C stereo microscope. Morphological characters of the holotype of *Burmaheptagenia zhouchangfai* were studied and interpreted based on images from the original description (*Chen & Zheng, 2023*). We processed the images using graphic tools embedded in Windows10 and 11.

## Terminology and taxonomic notes

General anatomical terminology is based on *Kluge (2004)* and *Bauernfeind & Soldán (2012)*. Abbreviations for wing veins used throughout the text follow *Kluge (2004)*, with some modifications by *Bauernfeind & Soldán (2012)*, *Staniczek, Godunko & Kluge (2018)*, and *Godunko, Martynov & Staniczek (2021)*. Morphological terms and abbreviations to describe thorax morphology used herein follow *Kluge (2004)*.

Taxonomic notes on previously described fossil species of Vietnamellidae are based on examination of the type material in the State Museum of Natural History Stuttgart (SMNS) and published data (see above).

## Nomenclatural acts

The electronic version of this article conforms to the requirements of the International Code of Zoological Nomenclature, and hence the new names contained herein are available under that Code from the electronic edition of this article. This published work and the nomenclatural acts it contains have been registered in ZooBank, the online registration system for the ICZN. The ZooBank LSIDs (Life Science Identifiers) can be resolved, and the associated information viewed through any standard web browser by appending the LSID to the prefix "http://zoobank.org/". The LSID for this publication is urn:lsid:zoobank.org:pub:B1FAF587-9EB2-4CDB-BD71-5AE0C9C6AD78. The new species is registered under urn:lsid:zoobank.org:act:6B45EB0E-8DF7-49D8-B317-E564009C00DC.

The online version of this work is archived and available from the following digital repositories: PeerJ, PubMed Central SCIE and CLOCKSS.

## RESULTS

### Systematic palaeontology

Subphylum Hexapoda Latreille, 1825
Class Insecta Linnaeus, 1758
Order Ephemeroptera Hyatt & Arms, 1890
Family Vietnamellidae Allen, 1984

***Burmella*** Godunko, Martynov & Staniczek, 2021

*Burmella* Godunko, Martynov & Staniczek, 2021 (in *Godunko, Martynov & Staniczek, 2021*: *ZooKeys*, 1036: 101)
= *Burmaheptagenia* Chen & Zheng, 2023 **syn. nov.** (in *Chen & Zheng, 2023*: *Cretaceous Research*, 151: 4)

**Type species.** *Burmella paucivenosa* Godunko, Martynov & Staniczek, 2021; ibid.: 101 (original designation)
**Species composition.** *Burmella paucivenosa* Godunko, Martynov & Staniczek, 2021 (male imago, SMNS BU–179); *Burmaheptagenia zhouchangfai* (Chen & Zheng, 2023) **comb. nov.** (male imago, CZT-EPH-MA5); *Burmella clypeata* Godunko, Martynov & Staniczek,
**Table 1** Measurements of the holotype of *Burmella inconspicua* sp. nov. (female imago; SMNS BU–385).

| Characters | mm | Characters | mm |
|---|---|---|---|
| Length of body | 5.62* | Length of tibia | 1.38 |
| Length of right foreleg | 2.93 | Length of tarsus | 0.48* |
| Length of femur | 0.82 | Segment I | 0.10 |
| Length of tibia | 1.30 | Segment II | 0.08 |
| Length of tarsus | 0.81 | Segment III | 0.08 |
| Segment I | 0.20 | Segment IV | 0.10 |
| Segment II | 0.15 | Segment V | 0.12* |
| Segment III | 0.14 | Length of right hind leg | 2.65 |
| Segment IV | 0.15 | Length of femur | 1.26 |
| Segment V | 0.17 | Length of tibia | 0.95 |
| Length of left foreleg | 2.89 | Length of tarsus | 0.44 |
| Length of femur | 0.80 | Segment I | 0.09 |
| Length of tibia | 1.27 | Segment II | 0.08 |
| Length of tarsus | 0.82 | Segment III | 0.08 |
| Segment I | 0.18 | Segment IV | 0.09 |
| Segment II | 0.15 | Segment V | 0.11 |
| Segment III | 0.14 | Length of left hind leg | 2.43* |
| Segment IV | 0.18 | Length of femur | 1.24 |
| Segment V | 0.17 | Length of tibia | 0.88* |
| Length of right middle leg | 3.34 | Length of tarsus | 0.31 |
| Length of femur | 1.42 | Segment I | 0.07 |
| Length of tibia | 1.37 | Segment II | 0.05 |
| Length of tarsus | 0.55 | Segment III | 0.05 |
| Segment I | 0.12 | Segment IV | 0.07 |
| Segment II | 0.10 | Segment V | 0.09 |
| Segment III | 0.10 | Length of right forewing | 5.45 |
| Segment IV | 0.11 | Length of left forewing | 5.05* |
| Segment V | 0.15 | Length of right hind wing | 0.53 |
| Length of left middle leg | 3.26* | Length of left hind wing | 0.50* |
| Length of femur | 1.40 | Hind/Fore wings length ratio | 0.10 |

**Notes.**
*Preserved part.

2021 (female imago, SMNS BU–321); *Burmella inconspicua* Godunko & Staniczek **sp. nov.** (female imago, SMNS BU–385).

**Locality and horizon.** Hukawng Valley, Kachin State, Myanmar (Burma); Cenomanian, mid-Cretaceous.

**Amended diagnosis of adults** (modified from *Godunko, Martynov & Staniczek, 2021*; see also Tables 1 and 2). Based on the study of the extinct taxa, as well as comparative analysis of extant *Vietnamella*, we provide here additional generic characters.

*Measurements* (**i**) Body length 5.75–7.00 mm; forewing length 4.64–5.45 mm; hind wing length 0.45–0.68 mm; hind wing very small, as long as 0.08–0.14 of forewing length;

*Male head* (**ii**) lower portion of compound eyes small and narrow;
**Table 2 The summary of morphological characters of the adults to distinguish representatives of extant *Vietnamella* and extinct *Burmella*.** Distinct generic differential characters are marked in bold.

| Characters | *Burmella paucivenosa* Godunko, Martynov & Staniczek, 2021 | *Burmella zhouchangfai* (Chen & Zheng, 2023) comb. nov. | *Burmella clypeata* Godunko, Martynov & Staniczek, 2021 | *Burmella inconspicua* sp. nov. Godunko & Staniczek | *Vietnamella* spp. |
|---|---|---|---|---|---|
| | Extinct; mid-Cretaceous Burmese amber, Upper Albian, max. age is 98.79 ± 0.62 Ma | | | | Extant; Indomalayan Realm |
| Adult (sex) | male imago | male imago | female imago | female imago | males \| females |
| *Measurements* | | | | | |
| **Body length (mm)** | 5.75 | 6.00 | 7.00 | 5.62[*] | 10–16 \| 12–14 |
| **Forewings length (mm)** | 4.64–4.68 | 5.00 | 5.12 | 5.05*–5.45 | 11–16 \| 12–15 |
| **Hind wings length (mm)** | 0.64–0.68 | 0.56 | 0.45 | 0.50*–0.53 | 2.55–3.20 \| 3.00–3.60 |
| **Hind/Forewings length ratio** | 0.14 | 0.11 | 0.08 | 0.10 | 0.20–0.25 |
| Forewings (width/length ratio) | 0.40 | 0.41 | 0.30[*] | 0.36 | 0.34–0.36 \| 0.34–0.37 |
| Hind wings (width/length ratio) | 0.94 | 0.65[*] | 1.00[*] | 0.80 | 0.72–0.86 |
| *Head* | | | | | |
| Eyes (shape and structure) | large, widely rounded, medially contiguous | large, widely rounded, not contiguous | elongated, large, widely separated medially, covered by anterolaterally expanded clypeal shield | large, moderately rounded, widely separated medially, uncovered by clypeal shield | large, widely rounded, not contiguous \| widely separated medially |
| **Eyes of males (lower portion)** | small, narrow | small, narrow | – | – | large, wide |
| Eyes of females (distance between inner margin of eyes/width of head ratio) | – | – | 0.73 | 0.57 | 0.52–0.60 |
| Clypeus in female imago (shape) | – | – | expanded antero-laterally, partly covering of eyes anteriorly | relatively small without expansions | relatively small without expansions |
| Facial keel in male imago (shape) | relatively small | relatively large | – | – | relatively small |
| *Thorax* | | | | | |
| Mesonotum | relatively narrow | relatively narrow | relatively narrow | – | relatively wide |

**Table 2** (*continued*)

| Characters | *Burmella paucivenosa* Godunko, Martynov & Staniczek, 2021 | *Burmella zhouchangfai* (Chen & Zheng, 2023) comb. nov. | *Burmella clypeata* Godunko, Martynov & Staniczek, 2021 | *Burmella inconspicua* sp. nov. Godunko & Staniczek | *Vietnamella* spp. |
|---|---|---|---|---|---|
| ANp (shape) | large, tapered and rounded anteriorly | ? large, tapered and rounded anteriorly | large, tapered and rounded anteriorly | – | large, nearly rectangular, broadly rounded anteriorly |
| **MS (distally)** | large | large | large | – | not large, narrow |
| **BS (shape)** | slightly elongated, trapezoidal | slightly elongated, trapezoidal | slightly elongated, trapezoidal | slightly elongated, trapezoidal | not elongated, nearly rectangular or slightly trapezoidal |
| FSp (inner margins) | subparallel | slightly tapered anteriorly or ? subparallel | subparallel | subparallel | subparallel or slightly expanded anteriorly |
| *Forewing* | | | | | |
| **Pterostigma (number of cross veins)** | 4–5 | at least 8 | – | 3 | 10–16 \| 9–17 |
| Pterostigma (shape of veins) | simple | simple | – | simple | simple and forked or mainly forked |
| Sc–RA (number of cross veins) | 8 | 10 | – | 3 | 8–12 \| 8–17 |
| R sector (number of free small intercalaries) | 1 | 1 | 5 | – | 1 |
| RS furcation (respectively to vein length) | 0.14 | 0.16 | at least 0.12 | 0.15 | 0.18–0.22 |
| iRS/RSp (number of cross veins) | 5 | 6 | at least 5 | 8 | 7–13 |
| iRS/RSa$_1$ (location veins basally) | not approximated | not approximated | ? approximated | approximated | approximated |
| RSp–MA$_1$ (number of basally connected / small free intercalaries) | – | 0/1 | 0/1 | – | 1/1 |
| MA furcation (respectively to vein length) | 0.60–0.62 | at least 0.57 | at least 0.50 | 0.55 | 0.53–0.58 |
| MA furcation (shape) | slightly asymmetrical | slightly asymmetrical | strongly asymmetrical | strongly asymmetrical | slightly asymmetrical to nearly symmetrical |
| MA$_1$/iMA/MA$_2$ (number of cross veins) | 2/1 | 3/2 | 1/2 | 3/2 | up to 3/5 |
| MA sector (number of basally connected / small free intercalaries) | 1/0 | 0/1 | 1/0 | – | up to 2/1 |

| Characters | *Burmella paucivenosa* Godunko, Martynov & Staniczek, 2021 | *Burmella zhouchangfai* (Chen & Zheng, 2023) comb. nov. | *Burmella clypeata* Godunko, Martynov & Staniczek, 2021 | *Burmella inconspicua* sp. nov. Godunko & Staniczek | *Vietnamella* spp. |
|---|---|---|---|---|---|
| $MA_2$–$MP_1$ (number of basally connected/small free intercalaries) | – | – | 1/0 | – | 1/0 |
| **MP sector (number of basally connected/small free intercalaries)** | 2/1 | 2/0 | 2/0 | 1/2 | up to 7/4 |
| **$MP_2$–CuA (number of intercalaries)** | one basally connected | one basally connected | – | one basally connected | up to 5 (2–3 basally connected; 1–2 free) |
| **Cubital sector (number of secondary bifurcate veins)** | 4 | 4 | 4 | 4 | 6 |
| Cubital sector (number of free small intercalaries) | – | at least 1 | at least 3 | 1 | 1–3 |
| **CuA–cubital basally connected intercalaries (number and location of cross veins)** | – | – | – | 1 (CuA–$iCu_1$) | 1 (CuA–$iCu_1$) or 1 (CuA–$iCu_3$) or 2 (CuA–$iCu_3$ and CuA–$iCu_5$) or 2 ($iCu_3$–$iCu_2$ and CuA–$iCu_5$) |
| **CuP (shape)** | smoothly curved towards wing base | smoothly curved towards wing base | smoothly curved towards wing base | smoothly curved towards wing base | sharply curved towards wing base |
| CuP–$A_1$ (number of free small intercalaries) | – | – | – | 1 | 1–2 |
| $A_1$–$A_2$ (number of cross veins) | – | 1 | – | – | – |
| $A_2$ (location basally) | arises from $A_1$ | closely approximated to $A_1$ | ? arises from $A_1$ | arises from $A_1$ | closely approximated to $A_1$ |
| *Hind wing* | | | | | |
| Wings (shape) | strongly rounded | slightly elongated | strongly rounded | rounded | rounded or slightly elongated |
| Costal process (shape) | rounded apically, markedly protruding | rounded apically, markedly protruding | rounded apically, shallow | rounded apically, markedly protruding | not protruding, distinctly shallow |
| Costal process (location to hind wing length) | in the middle | distinctly proximally | in the middle | slightly distally | distinctly proximally |
| Vein triads (number including secondary triads) | 1 | 1 | 1 | 1 | 1–3 |

**Table 2** (*continued*)

| Characters | *Burmella paucivenosa* Godunko, Martynov & Staniczek, 2021 | *Burmella zhouchangfai* (Chen & Zheng, 2023) comb. nov. | *Burmella clypeata* Godunko, Martynov & Staniczek, 2021 | *Burmella inconspicua* sp. nov. Godunko & Staniczek | *Vietnamella* spp. |
|---|---|---|---|---|---|
| **C–Sc [number of cross veins]** | 3 | – | 2 | 3 | 12–18 |
| **Sc–RA (number of cross veins)** | 2 | 2 | 2 | 3 | 8–15 \| 11–12 |
| RSa (location basally) | not connected | arises from RSp | arises from RSp | not connected | arises from RSp |
| **RA–RS (number of cross veins)** | ? 1 | 2 | – | – | 5–8 |
| RA–MA furcation (location) | present | present | present | absent | present |
| **RSp–MA (number of cross veins)** | – | – | – | – | 1–5 |
| **MA–MP (number of cross veins)** | – | 1 | – | – | 3–7 \| 5–7 |
| **MP–CuA (number of intercalaries)** | – | – | – | – | 1–5 |
| **MP–CuA (number of cross veins)** | – | ? 1 | – | – | 2–4 |
| **CuA–CuP (number of intercalaries)** | – | ? 1 | – | – | 2–4 |
| **CuA–CuP (number of cross veins)** | – | – | – | – | 1–4 |
| **CuP–A$_1$ (number of cross veins)** | – | – | – | – | 1–2 |
| *Abdomen* | | | | | |
| **Sternum VII (shape of subgenital plate in female)** | – | – | wide, distinctly convex | relatively narrow, distinctly convex | small, straight or slightly convex |
| **Sternum IX (shape of subanal plate in female)** | – | – | triangular, tapered and rounded apically | – | trapezoidal, tapered and truncated apically, or with shallow median cleft |
| Paracercus | absent | absent | absent | – | present |
| *Genitalia (male imago)* | | | | | |
| **Styliger plate (shape of median projection)** | large, strongly convex, widely rounded | large, strongly convex, widely rounded | – | – | straight or slightly convex |

(*continued on next page*)

**Table 2** (*continued*)

| Characters | *Burmella paucivenosa* Godunko, Martynov & Staniczek, 2021 | *Burmella zhouchangfai* (Chen & Zheng, 2023) comb. nov. | *Burmella clypeata* Godunko, Martynov & Staniczek, 2021 | *Burmella inconspicua* sp. nov. Godunko & Staniczek | *Vietnamella* spp. |
|---|---|---|---|---|---|
| **Gonostylus (length ratio of distal segments I–III)** | 1.00/0.20/0.18 | 1.00/0.16/0.10 | – | – | 1/0.75–0.80/ 0.12–0.14 |
| **Gonostylus distal segment III (shape)** | elongated | elongated | – | – | short, nearly round |
| **Gonostylus pedestal (shape)** | elongated, trapezoidal | elongated, trapezoidal | – | – | short, nearly square |
| **Penis lobes (shape)** | widely separated by V-shaped cleft, nearly tube-like | widely separated by V-shaped cleft, trapezoidal, tip strongly bent dorsally | – | – | fused, stout or slender, deep or shallow apicomedian cleft |

**Notes.**
*As preserved.

*Female head* (**iii**) anterior part of eyes covered or uncovered by anterolaterally expanded clypeal shield;

*Thorax* (**iv**) mesonotum relatively narrow; medioscutum markedly extended distally; basisternum of mesothorax nearly rectangular or slightly trapezoidal, slightly elongated;

*Forewings* (**v**) with small number of cross veins; pterostigma with less than nine simple veins, not anastomosed; MP sector with small number of marginal intercalaries; only one intercalary vein between $MP_2$ and CuA; two secondary bifurcate veins in cubital sector, mostly not connected by additional cross veins; CuP smoothly curved towards wing base; free marginal intercalaries along ventral margin are distributed in all vein sectors;

*Hind wings* (**vi**) strongly rounded; distinct reduction of the number of cross veins; RS triad present; MA, MP and secondary CuA and CuP triads absent; costal process developed, rounded apically, situated centrally or slightly distally;

*Abdomen* (**vii**) with vestigial gill sockets recognizable at least on segments II–VI; no trace of paracercus;

*Female genitalia* (**viii**) subgenital plate well developed, distinctly convex, widely rounded; subanal plate triangular-shaped, rounded apically;

*Male genitalia* (**ix**) with large median projection of styliger plate, widely rounded apically; three distal segments of forceps strongly elongated and slender; segment II longest, 5–6× as long as segment III; segments III and IV approximately of equal length or slightly longer; segment IV elongated; penis lobes widely separated by V-shaped cleft, tube-like or trapezoidal.

*Burmella inconspicua* Godunko & Staniczek sp. nov.

Figs. 1–4, Tables 1 and 2

**Material examined.** *Holotype.* Female imago in mid-Cretaceous Burmese amber, housed at SMNS under inventory number BU–385. Well preserved specimen, embedded in dorsoventral aspect (Figs. 1A, 1B). Wings almost entirely preserved, except of distal parts of veins C, Sc, RA and RSa in the left forewing (Figs. 1A, 1B, 2A, 2B). Dorsal side of thorax and abdominal tergites I–VII are damaged or lost. Left forewing is twisted and bent approximately at half of its length (Figs. 1A, 1C). Tarsomere 5 of left middle tarsus is damaged. Abdominal segments I–VII are complete; sterna VIII and IX are damaged; segment X and cerci not preserved (Figs. 1A, 1B).

**Derivation of name.** The species epithet is derived from the Latin adjective inconspicuus for its inconspicuous habitus.

**Diagnosis.** *Female imago.* Body length 5.62 mm (preserved part); *eyes* rounded, widely separated medially, not covered by a clypeal shield; distance between eyes $0.57\times$ of head width; *forewings* with four small, basally free marginal intercalaries in MP and Cu sectors, and between CuP and $A_1$; no small free marginal intercalaries in R sector; *hind wing* rounded, small, $0.10\times$ of forewing length; three cross veins between C–Sc; three cross veins between Sc–RA; no RS and RA–MA forks; *subgenital plate* approximately $0.15\times$ as wide as long, strongly convex, outer margin widely rounded.

**Description.** General colouration of body pale, light brown to brown; legs darker, brown to dark brown; body covered by irregular blackish maculation (Figs. 1A–1C).

Measurements: body length 5.62 mm (preserved part); forewing length up to 5.45 mm. Maximum forewing width $0.37\times$ maximum length. For complete measurements see Table 1.

*Head.* General colour brown; intensively brown maculation on eyes and facial keel. Clypeus not expanded, small; clypeal shield not developed. Antennae brown, slightly longer than head; flagellum slightly paler than scape and pedicel. Eyes brown, moderately rounded, widely separated medially; distance between eyes $0.57\times$ of head width; facets of eyes hexagonal. Ocelli large, without conspicuous colouration (Figs. 3A, 3C).

*Thorax.* General colouration of ventral cuticle light brown, lateral sides of terga paler; dorsal side of thorax missing. Prosternum relatively wide, light brown. Lateral aspect of thorax hardly visible. Mesosternum complete, light brown coloured; basisternum slightly elongated, nearly trapezoidal; furcasternal protuberances distinctly separated, with parallel inner margins. Metasternum relatively short (Fig. 1C).

*Forewings.* Hyaline, translucent, relatively narrow; venation well recognizable on right forewing and less visible on left forewing due to wing damage and deformation; three cross veins between Sc and RA, cross venation in other sectors well developed; no naturally pigmented vein sections between C and RA; no pigmented clouds around cross veins; no jagged edge along costal, tornoapical and basitornal margins. Pterostigma with three simple veins (Table 2; Figs. 2A, 2B).

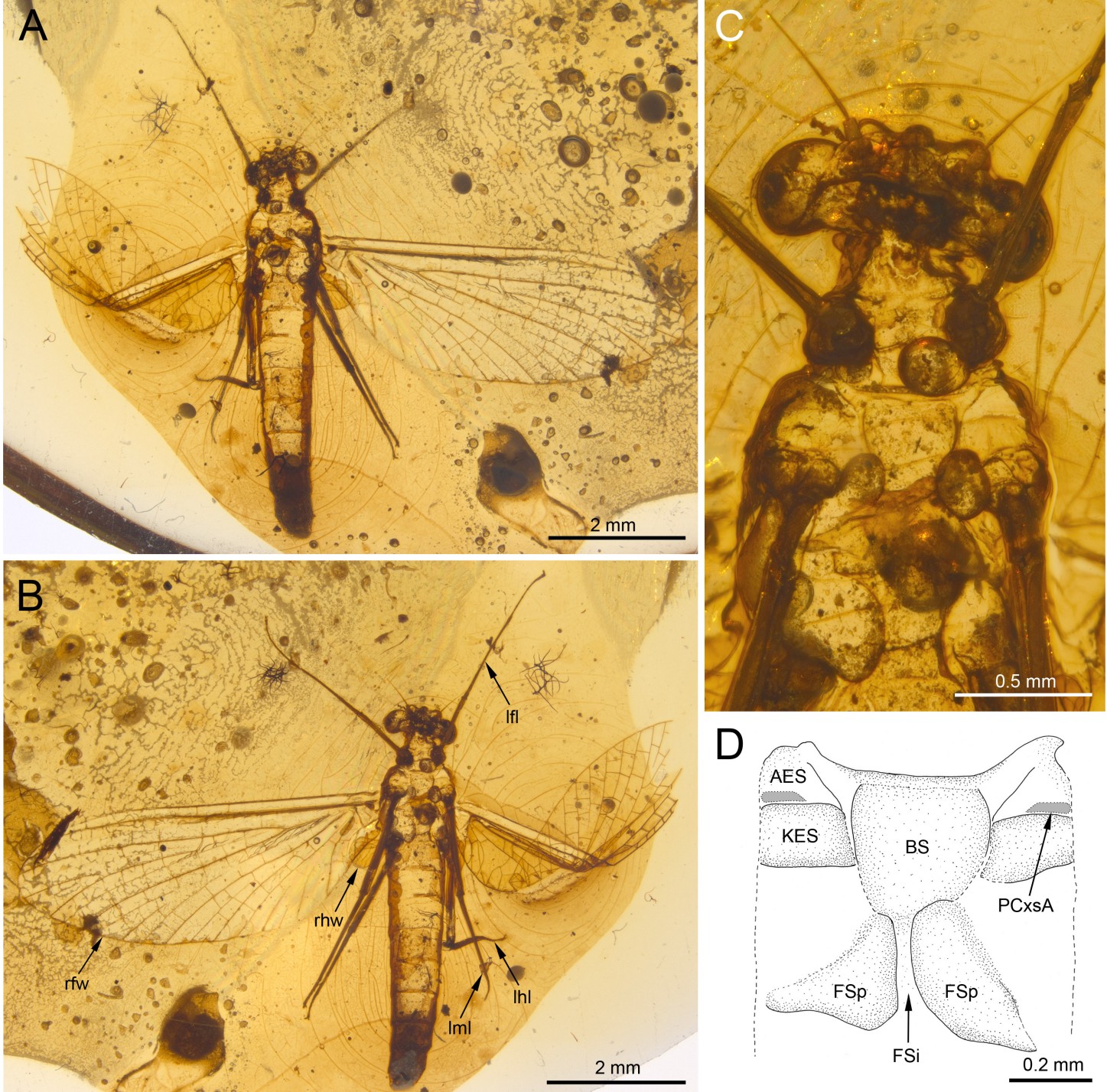

**Figure 1** ***Burmella inconspicua* sp. nov. (Vietnamellidae), mid-Cretaceous Burmese amber, holotype, female imago.** (A) Body in dorsal view, (B) body in ventral view, (C) head and thorax in ventral view, (D) mesosternum. Abbreviations: lfl, left foreleg; lhl, left hind leg; lml, left middle leg; rfw, right forewing; rhw, right hind wing. *Thorax* (*mesosternum*): AES, anepisternum; BS, basisternum; FSi, furcasternal impression; FSp, furcasternal protuberances; KES, katepisternum; PCxsA, anterior paracoxal suture. Scale bars: 2.0 mm (A, B), 0.5 mm (C), 0.2 mm (D).

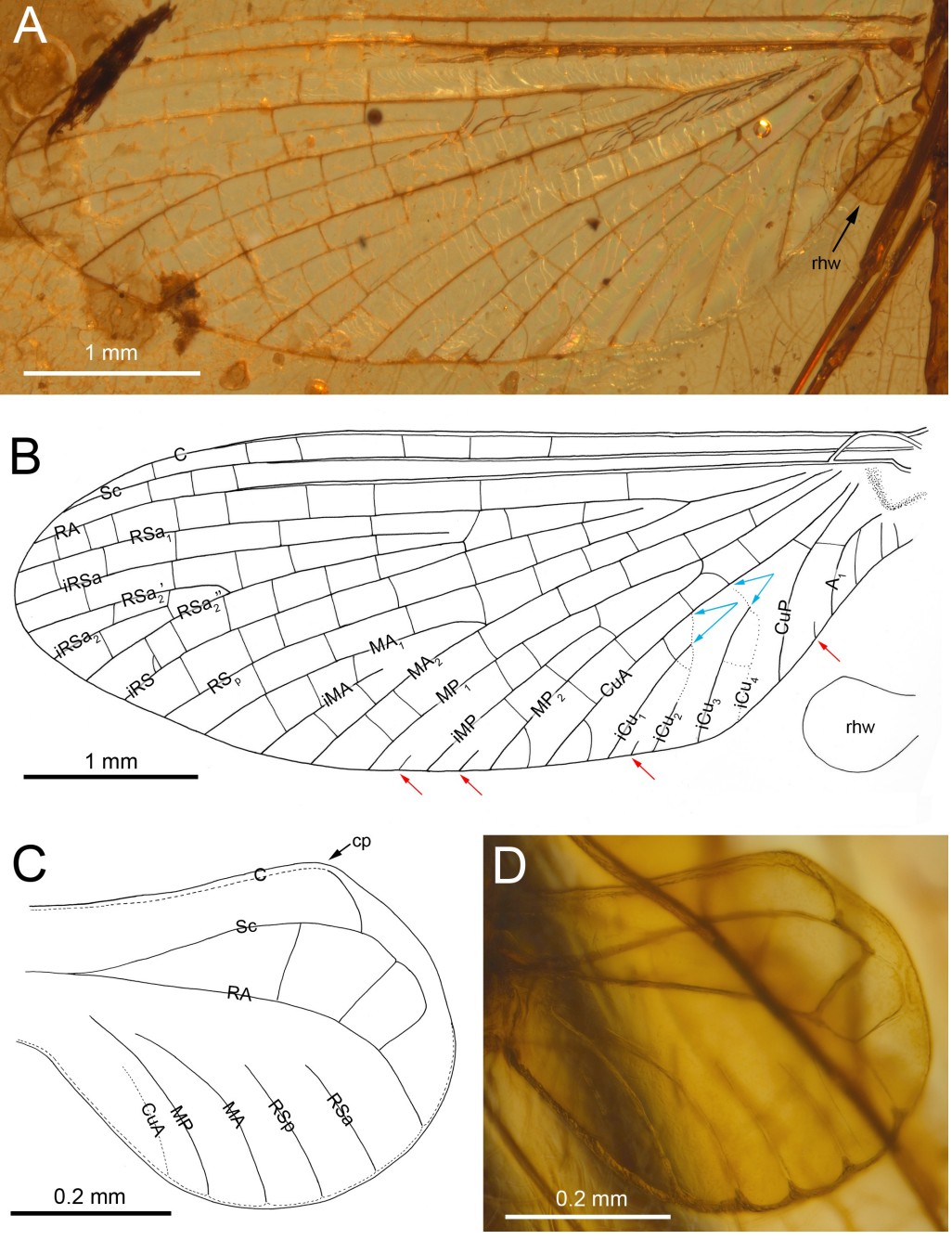

**Figure 2** *Burmella inconspicua* **sp. nov. (Vietnamellidae), mid-Cretaceous Burmese amber, holotype, female imago, wings.** (A) Right forewing in ventral view, (B) right forewing venation and size ratio of fore and hind wings, (C, D) right hind wing in dorsal view. Abbreviations: cp, costal process; rhw, right hind wing. Red arrows show basally free intercalary veins. Blue arrows show proximal ends of cubital intercalaries iCu$_1$–iCu$_4$. Scale bars: 1.0 mm (A, B), 0.2 mm (C, D).

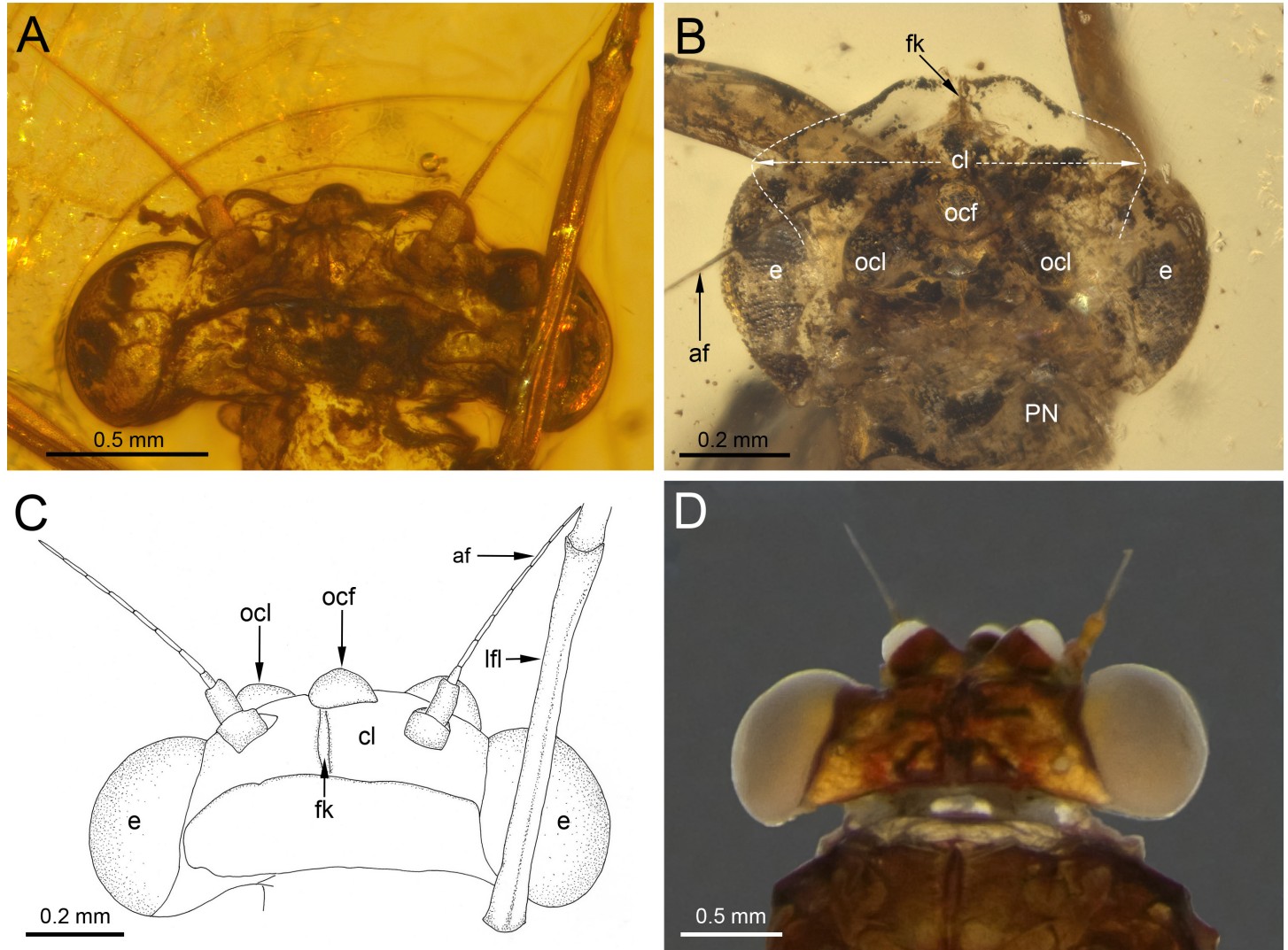

**Figure 3** **Heads of extinct and recent representatives of Vietnamellidae.** (A, C) *Burmella inconspicua* **sp. nov.,** mid-Cretaceous Burmese amber, holotype, female imago, (B) *Burmella clypeata* Godunko, Martynov & Staniczek, 2021, mid-Cretaceous Burmese amber, holotype, female imago, (D) *Vietnamella nanensis* Auychinda & Boonsoong, 2020, 1972, female imago (Thailand, Nan Province). Abbreviations: *Head and thorax*: af, antennal flagellum; cl, clypeus; fk, facial keel; e, eye; ocf, frontal ocellus; ocl, lateral ocelli; lfl, left foreleg; PN, pronotum. White dashed lines show laterally expanded clypeus in *B. clypeata*. Scale bars: 0.5 mm (A, D), 0.2 mm (B, C).

General pattern of forewing venation typical for *Burmella*: veins C and Sc light brown to brown; RS forked close near base, after 0.15 of its length; iRS long, connected with RSp by eight cross veins, approximated to $RSa_1$; MA fork strongly asymmetrical, forked after 0.55 of its length; iMA relatively short, $MA_1$ and $MA_2$ connected with iMA by 2–three cross veins; MP asymmetrical, forked after 0.40 of its length, $MP_1$ and $MP_2$ basally connected by a single cross vein; iMP relatively short, connected with $MP_1$ and $MP_2$ by three cross veins from each side; CuA closely approximated to CuP base; CuP smoothly curved toward wing base; *cua-cup* cross vein and *cup-a_1* cross vein present; *cua-cup* located distally from *cup-a_1*; two secondarily forked veins $iCu_{1+2}$ and $iCu_{3+4}$ arising from CuA to basitornal margin of

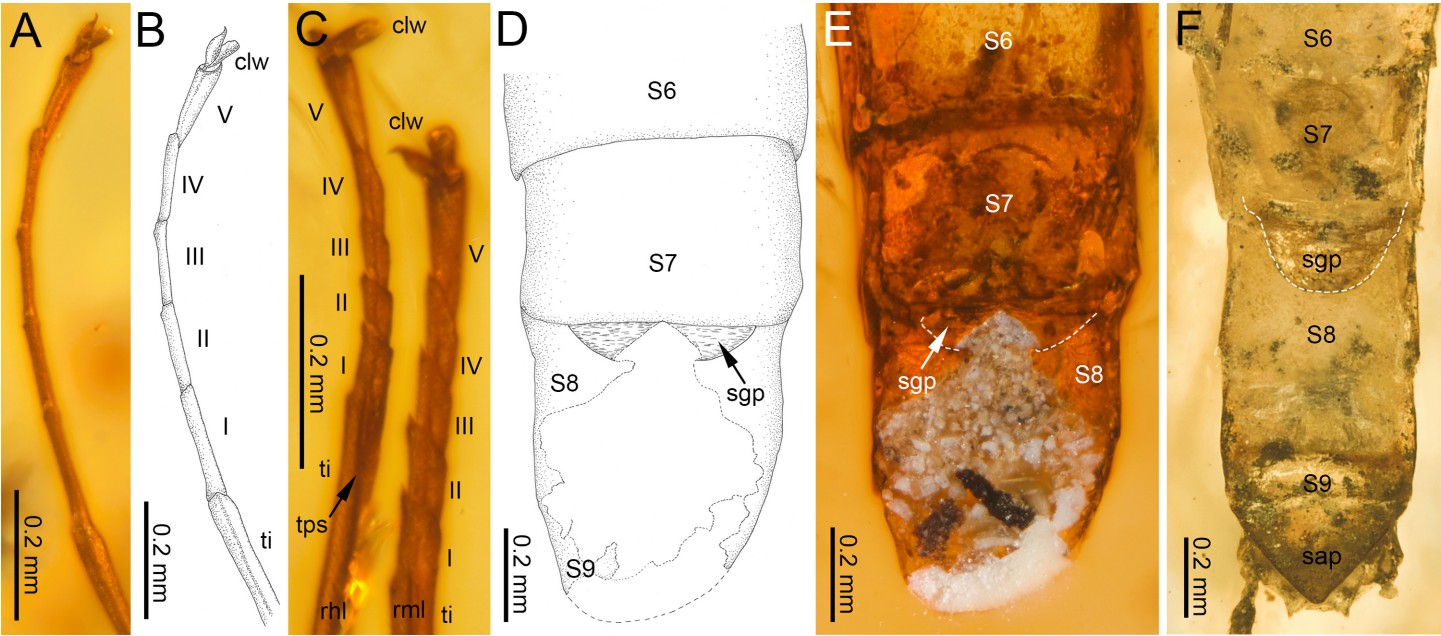

**Figure 4** Legs and end of female abdomen of *Burmella* Godunko, Martynov & Staniczek, 2021 from mid-Cretaceous Burmese amber. (A–E) *Burmella inconspicua* **sp. nov.**, mid-Cretaceous Burmese amber, holotype, female imago, (F) *B. clypeata* Godunko, Martynov & Staniczek, 2021, mid-Cretaceous Burmese amber, holotype, female imago. (A, B) Right foreleg, (C) right middle and hind legs, (D–F) tip of abdomen in ventral view. Abbreviations: *Legs* I–V, tarsomeres; tps, tibiopatellar suture; clw, tarsal claws; rml, right middle leg; rhl, right hind leg. *Abdomen*: S6-S9, abdominal sterna 6–9; sap, subanal plate; sgp, subgenital plate. White dashed lines show posterior margin of subgenital plate. Scale bars: 0.2 mm (A–F).

forewing (*i.e.,* in the cubital sector four veins $iCu_1$–$iCu_4$ each reaching basitornal margin); $iCu_1$ connected to CuA by additional cross vein. Vein $A_2$ arising from $A_1$; no cross veins in anal sector. Several intercalaries (*i.e.,* iRSa, $iRSa_2$, iMA, iMP) connected to longitudinal veins by 2–5 cross veins; four small, basally free marginal intercalaries in MP (two) and Cu (one) sectors, and between CuP and $A_1$ (one); one basally free marginal intercalary vein in cubital sector and one such vein between CuP and $MA_1$ (Table 2; Figs. 2A, 2B).

*Hind wings*. Hyaline, translucent, rounded, small, as long as $0.10\times$ of forewing length (Fig. 2B); venation light brown to brown; venation significantly reduced; no jagged edge along dorsal and ventral margins. Three cross veins between C–Sc and three cross veins between Sc–RA; no RS and RA–MA forks; RSa, RSp and MA short, not forked and not connected to other veins; MP longer than MA, not forked; no forks of cubital veins; CuA relatively short; CuP absent; no free small marginal intercalaries. Costal process rounded apically, protruding above anterior wing margin, situated slightly distally at middle of hind wing length (Table 2; Figs. 2C, 2D).

*Legs*. Almost preserved, except for damaged apical part of tarsomere V of left middle tarsus. Legs darker than body, brown to dark brown; no visible strong spines or setae on margins of leg segments. Tibiopatellar suture present on middle and hind legs, absent on forelegs (Figs. 4A–4C). First tarsomere of middle and hind legs fused with tibia. Forelegs: length ratio of femur/tibia/tarsus = 1/1.58/0.98; length ratio of tarsomeres:

1/0.79/0.74/0.84/0.90 (1 > 2 < 3 < 4 < 5). Middle legs: length ratio of femur/tibia/tarsus = 1/0.97/0.39; length ratio of tarsomeres: 1/0.95/0.95/1.11/1.58 (1 > 2 = 3 < 4 < 5) (Fig. 4C).

Hind legs: length ratio of femur/tibia/tarsus = 1/0.76/0.35 (proportion of tarsomere V calculated based on right leg only); length ratio of tarsomeres: 1/1.10/1.10/1.14/1.38 (1 > 2 = 3 < 4 < 5) (proportion of tarsomere V calculated based on right leg only). For details of measurements of tarsomeres see Table 1. Pretarsal claws dissimilar (one pad-like, the other one apically hooked) (Figs. 4A–4C).

*Abdomen.* Apical part of tergite X and most part of abdominal terga I–VII; preserved abdominal segments light brown to brown. Vestigial gill sockets hardly recognizable on segments V and VI; the other segments with gill sockets not distinguishable; no prominent posterolateral projections on preserved abdominal segments. Subgenital plate strongly convex, with presumably straight outer margin, relatively narrow, approximately 0.15× as wide as long. Subanal plate and cerci not preserved (Table 2; Figs. 4D, 4E).

**Affinities.** With the establishment of *B. inconspicua* **sp. nov.**, currently only two species of the genus *Burmella* are known based on adult females. The marked differences between these species are summarized in Table 2 and can be found especially in the head. While the eyes of *B. inconspicua* **sp. nov.** are large, widely separated, and anteriorly not covered by the clypeus, in *B. clypeata* the eyes are elongated, and the clypeus is expanded anterolaterally, covering the anterior part of eyes (Fig. 3B). As we have noted earlier (*Godunko, Martynov & Staniczek, 2021*), in contrast to all other known extant and fossil representatives of Vietnamellidae, only in *B. clypeata* the clypeus has such a characteristic shape (Fig. 3; compare with *e.g.*, *Auychinda, Sartori & Boonsoong, 2020*: p. 30, fig. 9A; *Auychinda et al., 2020*: p. 9, figs. 5A–5E).

While the venation pattern of the forewings is similar to other *Burmella* species, there are some minor differences in the number and location of free marginal intercalaries. In *B. inconspicua* **sp. nov.**, these are mainly distributed in MP and CuA sectors, whereas in *B. clypeata* there are more abundant free intercalary veins in R sector and between RSp and $MA_1$ (Table 2; *Godunko, Martynov & Staniczek, 2021*: p. 112, figs. 8D–8F).

Furthermore, in *Burmella*, the female forewings are slightly narrower than those of males, the width/length ratio of the well-preserved right forewing of *B. inconspicua* **sp. nov.** is 0.36. In males of *B. paucivenosa* and *B. zhouchangfai* the ratios are 0.40 and 0.41, respectively (Table 2; Figs. 5A, 5B). The female forewings of *B. clypeata* are partially damaged and deformed, but the left forewing, which is better preserved also appears relatively narrow, with a width/length ratio of 0.30 (as preserved) (for comparison see Table 2; *Godunko, Martynov & Staniczek, 2021*: p. 112, figs. 8C–8E; *Chen & Zheng, 2023*: p. 4: figs. 3A, 3B).

More differences can be found in the hind wings. The shape of hind wings of *B. clypeata* is poorly preserved due to fossilisation. Except for *B. zhouchangfai*, which has elongated hind wings bearing a basal costal process, the hind wings of other fossil Vietnamellidae are markedly rounded, with a costal process located at mid-length, as in the case of *B. inconspicua* **sp. nov.** slightly in the distal half. Additionally, in contrast to all extinct species, in *B. inconspicua* **sp. nov.** a R–MA furcation is absent, and the venation is generally very simplified.

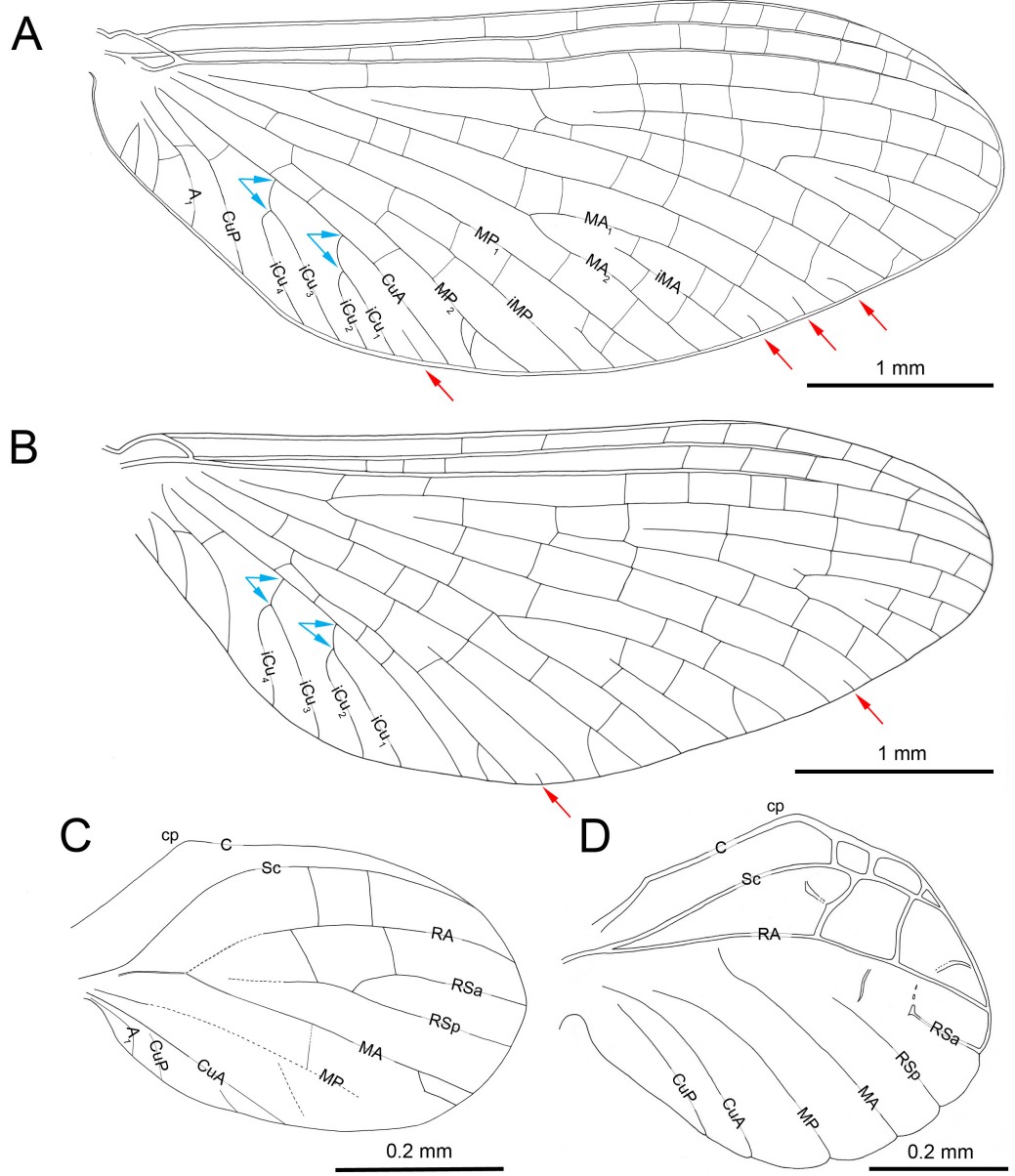

**Figure 5** **Wings of *Burmella* Godunko, Martynov & Staniczek, 2021 from mid-Cretaceous Burmese amber.** (A, C) *Burmella zhouchangfai* (Chen & Zheng, 2023), CZT-EPH-MA5, holotype, male imago, (B, D) *B. paucivenosa* Godunko, Martynov & Staniczek, 2021, holotype, male imago. (A) Left forewing, redrawn and interpreted from fig. 3A in *Chen & Zheng (2023)*, (B) right forewing, (C) left hind wing, redrawn and interpreted from fig. 3A in *Chen & Zheng (2023)*, (D) right hind wing. Abbreviations: cp, costal process. Red arrows show basally free intercalary veins. Blue arrows show proximal ends of cubital intercalaries iCu$_1$–iCu$_4$. Scale bars: 1.0 mm (A, B), 0.2 mm (C, D).

Finally, the proportions of the subgenital plate also differ, it is slightly wider in *B. clypeata* compared to the narrower plate in *B. inconspicua* **sp. nov.** (Table 2; Figs. 4D–4F).

As with *B. clypeata*, we cannot exclude that the female of *B. inconspicua* **sp. nov.** is conspecific to the male of *B. paucivenosa* or *B. zhouchangfai*. The preserved body part of

the female holotype of *B. inconspicua* **sp. nov.** is 5.62 mm in length, with the tenth segment of the abdomen lacking. Nevertheless, despite its incompleteness, the body length of this species is generally comparable to other representatives of the genus *Burmella*, which are distinctly smaller than extant Vietnamellidae. On the other hand, clear differences in the shape and venation of hind wings between all discovered fossil specimens rather point to the presence of four separate species. At least, as shown above, there is no doubt that females of *B. inconspicua* **sp. nov.** and *B. clypeata* belong to different species.

In order to maintain nomenclatural stability, we thus follow the approach to describe males and females of the same genus as different species, unless specimens of different sex are syninclusions and fossilised in mating position (see *e.g.*, *Staniczek & Godunko, 2016*; *Godunko, Neumann & Staniczek, 2019*; *Godunko, Martynov & Staniczek, 2021*).

## DISCUSSION

**Synonymy of the genera  *Burmella* = *Burmaheptagenia* syn. nov.**

***Burmella zhouchangfai*** **(Chen & Zheng, 2023) comb. nov.**

= *Burmaheptagenia zhouchangfai* Chen & Zheng, 2023 (in *Chen & Zheng (2023)*: Cretaceous Research, 151: 4, figs. 1–5)

The male imago of *Burmaheptagenia zhouchangfai* was described and illustrated in details by *Chen & Zheng (2023)*, with the same specific and generic diagnoses because of its assumed monotypic. The authors noted the presence of (**a**) separated (ca. dorsally) compound eyes; (**b**) median depression of furcasternum convergent anteriorly (*i.e.*, furcasternal protuberances not contiguous and separated); (**c**) rounded prosternum without any ridges; (**d**) numerous cross veins of forewings; (**e**) basal costal cross vein (*i.e.*, costal brace) strongly developed and attached anteriorly to costa; and (**f**) cubital sector with two pairs of intercalaries.

For the hind wings there was listed (**g**) a low number of cross veins and marginal intercalaries; and (**h**) MA forked apically.

Additionally, (**i**) the length ratio of femur, tibia and tarsus of all legs were given, together with information on the presence of (**j**) a four-segmented tarsus of middle and hind legs, (**k**) as well as all tarsal claws uniformly hooked.

Finally, *Chen & Zheng (2023)* mentioned (**l**) an enlarged forceps base (*i.e.*, gonostylus pedestal); (**m**) a three-segmented forceps, with a length ratio of approx. 9:2:1; (**n**) a bilobed, upcurved penis; and (**o**) a vestigial paracercus.

After summarizing the data in *Chen & Zheng (2023)* and comparing it with morphological characters studied in fossil species of *Burmella*, especially in *B. paucivenosa*, we place *Burmaheptagenia zhouchangfai* in the family Vietnamellidae, proposing generic synonymy between *Burmella* and *Burmaheptagenia* **syn. nov.** based on the following:

(**a**) The male compound eyes of *B. zhouchangfai* are well-developed, large and widely rounded, and medially separated. A clear separation of compound eyes into two portions is well recognizable laterally, with the lower portion being small and narrow (see fig. 2B in *Chen & Zheng, 2023*; Fig. 6A). The same structure of compound eyes is described for the male holotype of *B. paucivenosa* (Table 2; Fig. 6B). Moreover, a marked border between upper and lower portions is characteristic for males of all extant Vietnamellidae and does not occur in Heptageniidae (see *Godunko, Martynov & Staniczek, 2021*: p. 105, figs. 2B, 2C; *Auychinda et al., 2020*: p. 28, 29, figs. 7B, 8B; *Auychinda, Sartori & Boonsoong, 2020*, p. 8, figs. 4A, 4C).

(**b**) Depending on the location of the mesothoracic nerve ganglion, the position of the mesothoracic protuberances may differ among mayflies. In *Burmella* (as well as in all other Ephemerelloidea), they are well-separated (Fig. 7D), which is a non-unique apomorphy of Ephemerelloidea according to *Kluge (2004)*. The same character state is also present in many other mayfly taxa, including Heptagenioidea (*Kluge et al., 1995*). In contrast to most Heptagenioidea, in all taxa of Ephemerelloidea and Leptophlebioidea the mesothoracic ganglion is located in the anterior part of the furcasternum. Therefore, the median impression of the adult furcasternum has parallel sides or diverges anteriorly (*Kluge, 2004*). In extant Vietnamellidae, either character state can be present (Table 2; Fig. 7C; see also *Auychinda, Sartori & Boonsoong, 2020*: p. 28–30, figs. 7G, 8G, 9G; *Auychinda et al., 2020*, p. 8–9, figs. 4E, 5E). *Chen & Zheng (2023)* described the mesothoracic furcasternal protuberances in *B. zhouchangfai* as convergent anteriorly and the margins depicted by a dashed line (see Fig. 2B in the original description). It is however difficult to evaluate this character, because the specimen is embedded in lateral position, hampering the examination of the thoracic sterna. However, re-evaluating this character, we assume that these protuberances are parallel like in other species of *Burmella* rather than convergent. The same difficulties refer to the mesonotum, which is poorly visible due to its body position and the vertically folded forewings. However, from the characteristics visible in the original images, it can be concluded that the mesonotum is similar to the ones in *B. paucivenosa* and the remaining species of *Burmella*: (**i**) the mesonotum is relatively narrow, with the mesoscutum markedly extended distally; (**ii**) the mesonotal suture is transverse and distinctly expressed; (**iii**) the recognisable anterior part of the medioparapsidal suture is relatively straight; (**iv**) the lateroparapsidal suture is distinctly curved laterally, reaching the posterior scutal protuberance; (**v**) the scutellum is elongated and not modified otherwise (Figs. 6C, 7A). In extant Vietnamellidae, the mesonotum is relatively wider and shorter, but otherwise similar to those of fossil species (Fig. 7B; *Auychinda, Sartori & Boonsoong, 2020*; *Auychinda et al., 2020*).

(**c**) *Chen & Zheng (2023)* pointed out that the prosternum in *B. zhouchangfai* is rounded and without any ridges. Comparing the shape of prosterna

with Heptageniidae, the authors attributed *B. zhouchangfai*, indicated its close relationship to Rhithrogeninae and Ecdyonurinae, in contrast to Heptageniinae, which always have a distinct transversal ridge (*Kluge, 2004*: p. 171, fig. 56A). In any case, the absence of such a ridge in *B. zhouchangfai* agrees well with the structure of the prosternum in Vietnamellidae, which also has no specific ridges.

(**d, e**) The total number of forewing cross veins in *B. zhouchangfai* is comparable to many Heptageniidae. At the same time, the number of cross veins and other characteristics of forewing venation of this species are closer to representatives of the genus *Burmella* (especially *B. paucivenosa*) (Table 2). Also, in species of *Burmella*, the cross venation is less developed than in *Vietnamella* (see Diagnosis; Table 2; Figs. 2A, 2B, 5A, 5B; *Kluge, 2004*: p. 318, fig. 95A; *Godunko, Martynov & Staniczek, 2021*: p. 107, 112, figs. 4A, B, 8C, D; *Chen & Zheng, 2023*: p. 4, figs. 3A, B). The costal brace in all species of Vietnamellidae is anteriorly fused with the costa and thus typical for most species of Ephemeroptera (see also *Staniczek, Bechly & Godunko, 2011*).

(**f**) For all taxa of Vietnamellidae discussed here, the presence of up to three forked veins arising from CuA to the basitornal margin of forewing is characteristic. In *Burmella*, only two such veins are recorded in all known species, including *B. zhouchangfai* (*Kluge, 2004*; *Godunko, Martynov & Staniczek, 2021*; *Chen & Zheng, 2023*). In contrast to adults of Vietnamellidae with clearly bifurcate veins, the cubital intercalaries of both extinct and recent Heptageniidae are basally free, only connected by regular cross veins to CuA, CuP, basitornal margin, and between each other. Additionally, CuP in *B. zhouchangfai* is smoothly curved toward the hind margin of wing (in contrast to CuP of *Vietnamella*, which is sharply curved at approximately 1/3 of its length) (curved inwards according to *Jacobus & McCafferty, 2006*) (see also *Godunko, Martynov & Staniczek, 2021*).

(**g, h**) The hind wings of *B. zhouchangfai* have some specific features that are not found in any other species of the genus. *Chen & Zheng (2023*, fig. 3C) briefly described and depicted the hind wings. Our new interpretation of the hind wings (Fig. 5C) is somewhat different. We noticed the presence of an inconspicuous costal process, which is situated proximally near the wing base, rounded apically, and markedly protruding above the anterior wing margin (Fig. 5C). This resembles the condition in extant species of *Vietnamella*, in contrast to other representatives of *Burmella*, in which the costal process of the hind wing is situated at nearly half-length or slightly distally of the middle (see Table 2; Figs. 5C, 5D). Regarding the interpretation of the hind wing venation, we suggest that (**i**) RA–MA furcation is symmetrical, however RS does not arise from this fork as depicted by *Chen & Zheng (2023*, fig. 3C); (**ii**) RS is asymmetrically forked on RSa and RSp; (**iii**) MA is not forked, as in all other Vietnamellidae, instead an intercalary vein is basally attached to MA (erroneously indicated as a fork of MA by *Chen & Zheng, 2023*); (**iv**)

additionally, MP, CuA, CuP and putative A$_1$ veins are hardly visible. The hind wing length of *B. zhouchangfai* is similar to the ones in other species of *Burmella*, ranging between 0.45–0.68 mm in this genus (0.56 mm in *B. zhouchangfai*). Among all Vietnamellidae, only in *B. zhouchangfai* the hind wings have a roundish shape with moderately arched fore margin, slightly elongated, with a width/length ratio of 0.65 (as preserved), while in other species this ratio is 0.72–1.00, and more particularly 0.80–1.00 in the remaining species of *Burmella* (see Table 2). However, the interpretation of the hind wing shape and proportions in *B. zhouchangfai* remains difficult, as the wings of the male are not visible at right angle, but at a smaller angle. In addition, there is a resin layer along the posterior margin and at the base of the hind wing, which would distort its real proportions. The hind wings of *Burmella* are strongly reduced, as long as 0.08–0.14 of the forewing length (0.11 in *B. zhouchangfai*), in contrast to the extant species of Vietnamellidae, which have a ratio between 0.20 and 0.25 (see Table 2; *Godunko, Martynov & Staniczek, 2021*). Reduction or absence of the hind wings occurs in many mayfly taxa, and this character state is independently developed in several lineages of Baetoidea, Caenoidea, Ephemerelloidea and Leptophleboidea (*Kluge, 2004*). Within fossil mayflies these features are mentioned for a few Mesozoic and Cenozoic taxa. *Staniczek (2003)*, *Godunko & Krzemiński (2009)* and *Staniczek, Godunko & Krzemiński (2017)* reported the presence of very small hind wings in *Borinquena* Traver, 1938 from Miocene Dominican amber. Recently, *Chen & Zheng (2024)* described the monotypic genus *Crephlebia zhoui* Chen & Zheng, 2024 from Mid-Cretaceous Burmese amber (originally attributed to Leptophlebiidae), based on a male imago lacking hind wings.

**(i–k)** The length ratio of femur/tibia in male forelegs of *B. zhouchangfai* is similar in the male of *B. paucivenosa*, comprising approximately 1/2. Tarsomeres of the forelegs are not completely preserved in these two species and the shape of tarsal claws remains unknown. The structure and proportions of the middle and hind leg segments are similar in these species. In *B. paucivenosa*, the length ratio of femur/tibia/tarsus in the middle leg is 1/0.71/0.21 and in hind legs maximally 1/0.91/0.51 . In contrast, *B. zhouchangfai* in both legs has a length ratio of 1/0.8/0.3 in (for more details see *Godunko, Martynov & Staniczek, 2021*; *Chen & Zheng, 2023*). *Chen & Zheng (2023)* indicated the presence of only four tarsomeres in middle and hind legs. In fact, the number of tarsomeres in *B. zhouchangfai* is rather five as in other fossil and extant Vietnamellidae, with the first tarsomere of middle and hind legs short and fused with the tibia. In contrast to the information published by *Chen & Zheng (2023)* concerning the presence of hooked claws of the middle and hind legs, that the re-investigation rather points to an ephemeropteroid condition in all claws , with the outer claw hooked and the inner claw blunt (Figs. 6D–6F). Finally, a tibiopatellar suture is well visible on middle and

hind legs; the same condition is present not only in Heptageniidae, but partly also in Ephemerelloidea, including Vietnamellidae (Figs. 4C, 6D–6G).

(l–o)   The male genitalia of in both species of *Burmella* have a characteristic shape very different from that of extant *Vietnamella*. The styliger plates of *B. zhouchangfai* and *B. paucivenosa* are of similar shape, with large, median, apically widely rounded projection, markedly protruding above the anterior margin of styliger, in contrast to the straight or slightly convex central part of the styliger plate in *Vietnamella* (Figs. 8A–8C; *Auychinda, Sartori & Boonsoong, 2020*; *Auychinda et al., 2020*). Pedestals of gonostylus (see *Kluge, 2004*) of *Burmella* males are distinctly elongated, tapered apically, in contrast to distinctly short gonostylus pedestals in *Vietnamella*. Imaginal gonostyli of *B. zhouchangfai* and *B. paucivenosa* also have a similar shape, corresponding to the majority of Ephemerelloidea. Gonostyli are equipped with three distal segments, the first of which is by far the longest, the second segment is shorter, however still longer than the apical segment. The length ratio of gonostyli segments I–III in *Burmella* is 1/0.16–0.20/0.10–0.18, in contrast to 1/0.75–0.80/0.12–0.14 in *Vietnamella*. In the latter, the first segment is only slightly longer than the well-developed second segment; third segment is nearly round and markedly shorter than in other species of Vietnamellidae known on males (Table 2; Figs. 8A–8C; *Auychinda, Sartori & Boonsoong, 2020*; *Auychinda et al., 2020*). *Chen & Zheng (2023)* described the penis of *B. zhouchangfai* as "…bilobed, constricted at half-length; each lobe upcurved, with two elliptical sclerites laterally, apex pointed". A detailed examination of the genitalia of *B. zhouchangfai* indicates close similarity to *B. paucivenosa*. In both fossil species penis, the penis lobes are relatively simple and widely separated by a V-shaped cleft (Figs. 8A–8C). In contrast to *Burmella*, the penis lobes of *Vietnamella* are fused almost at entire length with a shallow median cleft apically (*Auychinda et al., 2020*, p. 28, figs. 7J–7O). Compared to the nearly straight penis lobes of *Vietnamella*, the males of *Burmella* have the tip of the penis bent dorsally, more pronounced in *B. zhouchangfai* than in *B. paucivenosa* (Figs. 8A–8C; *Godunko, Martynov & Staniczek, 2021*; *Chen & Zheng, 2023*). Finally, within Ephemerelloidea, apart from extant larvae and adults of *Dicercomyzon* Demoulin, 1954 and *Teloganodes* Eaton, 1882, only adults of *Burmella* have a reduced paracercus.

Additionally, judging from the image of *B. zhouchangfai* in lateral view (*Chen & Zheng, 2023*: p. 5, fig. 4E), a putative remnant of a gill socket can be faintly recognised at least on the left side of abdominal segment IV. The same character is reported for all fossil species of *Burmella*, including *B. inconspicua* **sp. nov.**, with the presence of recognizable gill sockets at least on abdominal segments II-VII (see also in *Godunko, Martynov & Staniczek, 2021*). In adults of *Vietnamella*, the gill sockets are present on abdominal segments I–VII, but always hardly recognizable on segment I. On the other hand, there are neither traces of gill sockets in fossil nor extant adult Heptageniidae. Moreover, in contrast to Heptageniidae,

in *Burmella* (including *B. zhouchangfai*) there are either free small intercalaries or basally connected intercalary veins present along the posterior margin of the forewings (Figs. 2A, 2B, 5A).

Finally, body and forewing size of *B. zhouchangfai* is similar to those of other species of *Burmella*. All are small mayflies with body and forewing length two or more times shorter than those of extant Vietnamellidae (see Table 2). Males of *Burmella* are smaller than females, at least the completely preserved body and forewing of *B. clypeata* is longer than in *B. paucivenosa* and *B. zhouchangfai*. The female imago of *B. inconspicua* **sp. nov.** was evidently also slightly larger than males, considering the size of forewing and preserved part of the body (see Table 2). All of these discussed characters, as well as the other features listed above, strongly suggest that *B. zhouchangfai* indeed belongs to the genus *Burmella* within the family Vietnamellidae.

## Differentiation between *B. paucivenosa* and *B. zhouchangfai*

By establishing synonymy between *Burmella* and *Burmaheptagenia*, we can also confirm that *B. paucivenosa* is well separated from *B. zhouchangfai*. The male of *B. zhouchangfai* is well distinguished from *B. paucivenosa* by: (**i**) the structure of compound eyes and size of facial keel; (**ii**) forewing venation with larger number of cross veins in pterostigma; location of basal end of $A_2$, and presence of cross veins in anal sector; number of marginal intercalaries; (**iii**) shape, proportions and venation of hind wings; and (**iv**) structure of gonostyli and penis lobes (Table 2).

**Amended diagnosis of male imaginal *Burmella paucivenosa*** (modified from *Godunko, Martynov & Staniczek, 2021*); see also Table 2; Figs. 6B, 6G, 5B, 5D, 8B, 8C): body length 5.75 mm; forewing length 4.64–4.68; *compound eyes* contiguous medially; *facial keel* relatively small; *forewings* with 4–5 cross veins in pterostigma; 3–4 marginal intercalaries connected with longitudinal veins, and two free small marginal intercalaries; $A_2$ arises from $A_1$; no cross veins in anal field; *hind wings* strongly rounded, small, $0.94\times$ as wide as long; costal process situated nearly at half-length of hind wing; three cross veins between C and Sc; RSa free, not arising from RSp; no marginal intercalaries; *gonostylus segment III* nearly equal to segment II in length; *penis lobes* obliquely truncate and moderately rounded apically, nearly tube-like; tip of penis lobes slightly bent dorsally; strong apical tooth on outer margin of lobes.

**Amended diagnosis of male imaginal *Burmella zhouchangfai*** (modified from *Chen & Zheng, 2023*); see also Table 2; Figs. 6A, 6C–6F, 5A, 5C, 8A): body length 6.00 mm; forewing length 7.00; *compound eyes* separated medially; *facial keel* relatively large; *forewings* with at least eight cross veins in pterostigma; up to two marginal intercalaries connected with longitudinal veins, and four free small marginal intercalaries; $A_2$ closely approximated to $A_1$; one cross vein in anal field; *hind wings* slightly elongated, small, at least $0.65\times$ as wide as long; costal process situated near to the wing base; no cross veins between C and Sc; RSa arises from RSp; at least two marginal intercalaries; *gonostylus segment III* shorter than segment II; *penis lobes* convergent apically, trapezoidal; tip of penis lobes obtuse, distinctly bent dorsally without any apical teeth or spines.

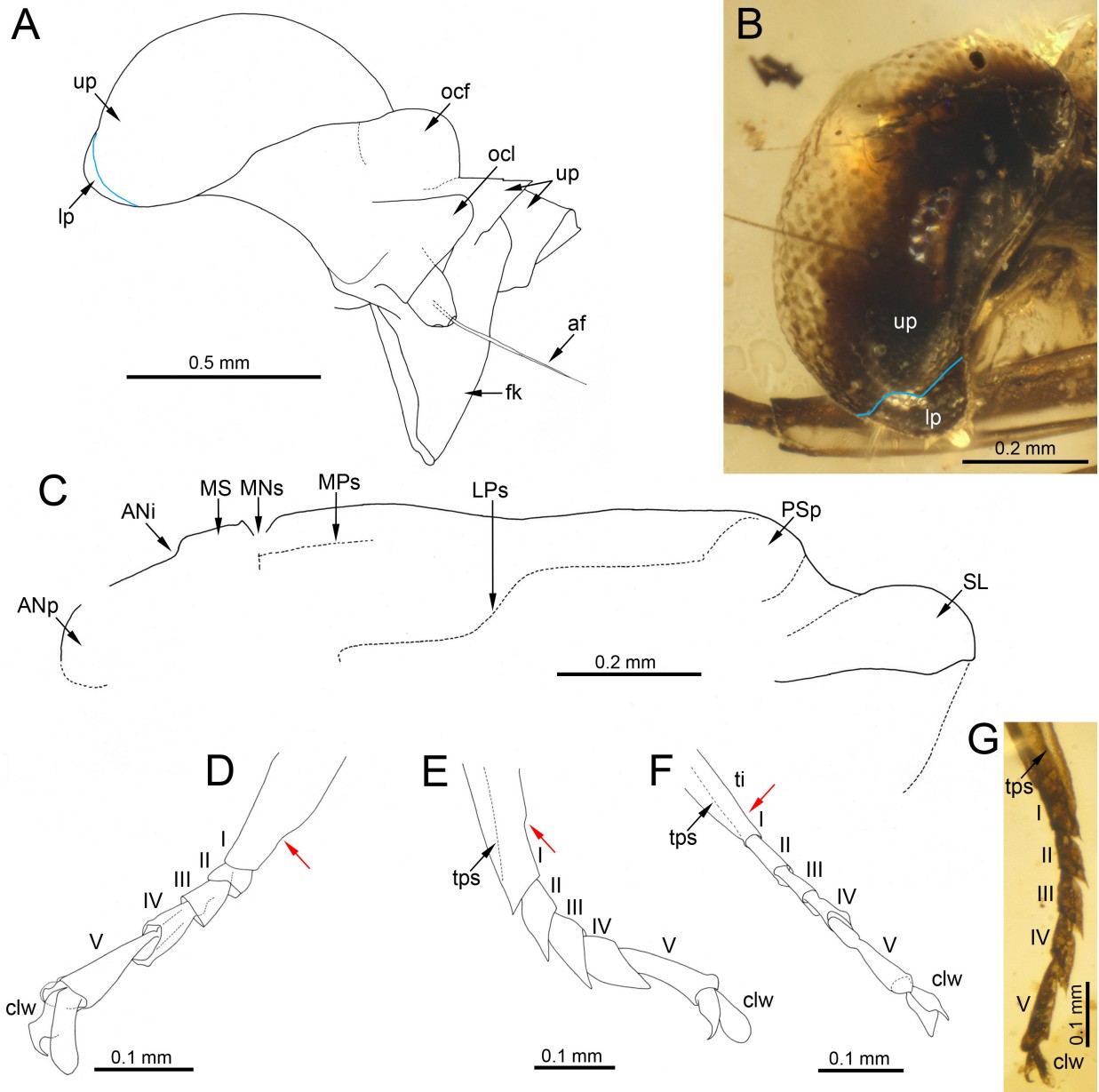

**Figure 6** **Head, thorax and legs of *Burmella* Godunko, Martynov & Staniczek, 2021 from mid-Cretaceous Burmese amber.** (A, C–F) *Burmella zhouchangfai* (Chen & Zheng, 2023), CZT-EPH-MA5, holotype, male imago, (B, G) *B. paucivenosa* Godunko, Martynov & Staniczek, 2021, holotype, male imago. (A) Head in right lateral view, redrawn and interpreted from fig. 2B in *Chen & Zheng (2023)*, (B) compound eye in left lateral view, (C) mesonotum in left lateral view, redrawn and interpreted from fig. 3A in *Chen & Zheng (2023)*, (D) tarsus of left middle leg, redrawn and interpreted from fig. 4D in *Chen & Zheng (2023)*, (E) tarsus of right hind leg, redrawn and interpreted from fig. 4E in *Chen & Zheng (2023)*, (F) tarsus of left hind leg, redrawn and interpreted from fig. 4F in *Chen & Zheng (2023)*, (G) tarsus of right middle leg. Abbreviations: *Head:* af –antennal flagellum, fk, facial keel; lp, lower portion of eye; ocl, lateral ocelli; ocf, frontal ocellus; up, upper portion of eye. Blue line shows the border between portions of compound eye. *Thorax*: ANi, anteronotal impression; ANp, anteronotal protuberance; LPs, lateroparapsidal suture; MNs, mesonotal suture; MPs, medioparapsidal suture; MS, medioscutum; PSp, posterior scutal protuberance; SL, scutellum. *Legs*: I–V, tarsal segments; tps, tibiopatellar suture; clw, tarsal claws. Red arrows show the proximal margin of tarsal segment I fused with the tibia. Scale bars: 0.5 mm (A), 0.2 mm (B, C), 0.1 mm (D, E, F, G).

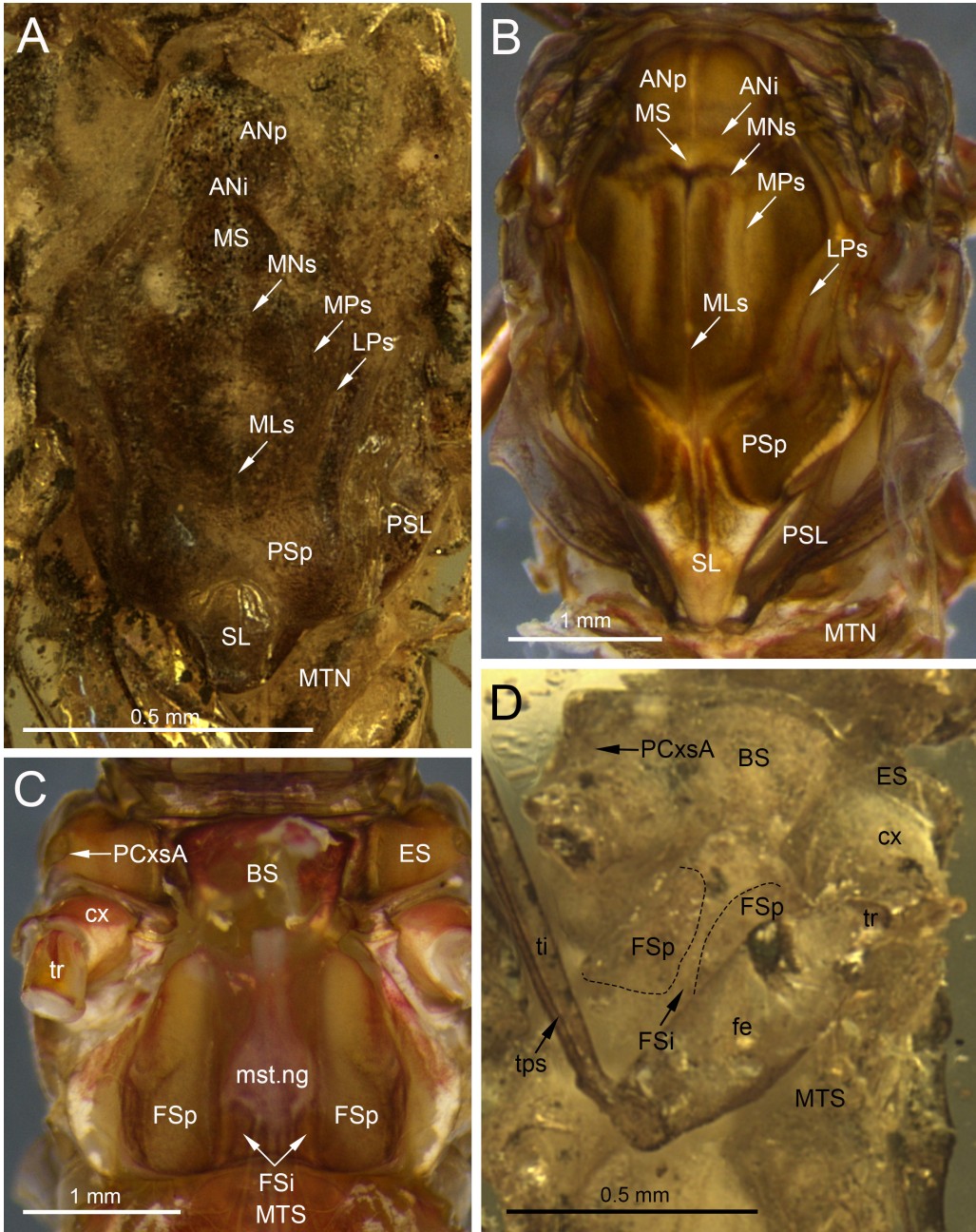

**Figure 7** **Thoracic structures of extinct and recent representatives of Vietnamellidae.** (A, D) *Burmella clypeata* Godunko, Martynov & Staniczek, 2021, holotype, female imago, (B, C) *Vietnamella nanensis* Auychinda & Boonsoong, 2020, 1972, female imago (Thailand, Nan Province). (A, B) Mesonotum, (C, D) mesosternum. Abbreviations: *Thorax (mesonotum)*: ANi, anteronotal impression; ANp, anteronotal protuberance; LPs, lateroparapsidal suture; MLs, median longitudinal suture; MNs, mesonotal suture; MPs, medioparapsidal suture; MS, medioscutum; MTN, metanotum; PSL, parascutellum; PSp, posterior scutal protuberances; SL, scutellum, (*mesosternum*): BS, basisternum; ES, episternum; FSi, furcasternal impression; FSp, furcasternal protuberances; (continued on next page…)

**Figure 7 (…continued)**
mst.ng, mesothoracic nerve ganglion; MTS, metasternum; PCxsA, anterior paracoxal suture. *Legs*: cx, coxa; fe, femur; ti, tibia; tr, trochanter; tps, tibiopatellar suture. Black dashed lines show the margins of furcasternal projections. Scale bars: 0.5 mm (A, D), 1.0 mm (B, C).

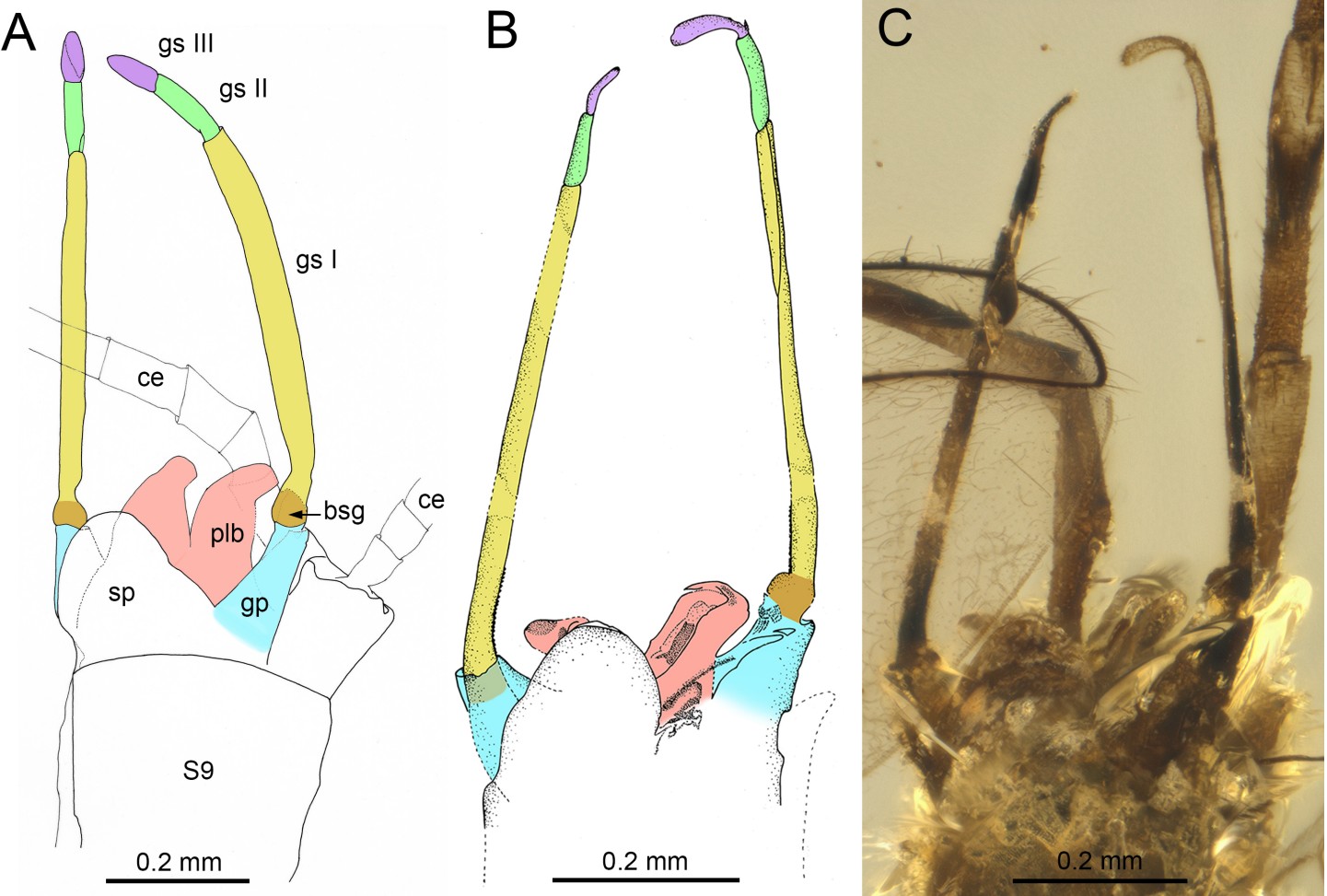

**Figure 8** **Genitalia of *Burmella* Godunko, Martynov & Staniczek, 2021 from mid-Cretaceous Burmese amber.** (A) *Burmella zhouchangfai* (Chen & Zheng, 2023), holotype, male imago, genitalia in ventrolateral view, redrawn and interpreted from fig. 5C in *Chen & Zheng (2023)*, (B, C) *B. paucivenosa* Godunko, Martynov & Staniczek, 2021, holotype, male imago, genitalia in ventrolateral view, based on *Godunko, Martynov & Staniczek (2021)*. Abbreviations: S9, abdominal sternum IX; ce, cercus; bsg, basal segment of gonostylus (light brown coloured); gp, gonostylus pedestal (blue coloured); gs I, gs II, gs III, distal gonostyli segments I (yellow coloured), II (green coloured) and III (violet coloured); plb, penis lobes (pink coloured); sp, styliger plate. Scale bars: 0.2 mm (A–C).

## Systematic placement of the genus *Burmella*

*Godunko, Martynov & Staniczek (2021)* noted that the general shape of male genitalia in *Burmella*, as well as the presence of a divided mesothoracic furcasternum in combination with two secondarily forked veins arising from CuA to the basitornal margin of forewing, resemble the combination of characters also found in Heptageniidae. This obviously led to the initial erroneous placement of *B. zhouchangfai* in the family Heptageniidae (*Chen &*

*Zheng, 2023*). However, the species here assigned to the genus *Burmella* also share the most important apomorphic characters of Ephemerelloidea (see also *Kluge, 2004*), namely in the forewing the presence of a transverse vein *cua-cup* localised more distally than *cup–a$_1$*; more than one bifurcated vein in the cubital sector of the forewing; the specific arrangement of thoracic sutures as discussed in detail above, and well separated mesothoracic furcasternal protuberances. The assignment of *Burmella* to Vietnamellidae within Ephemerelloidea is more difficult, as the family is mostly defined by larval apomorphies. However, it presents rounded hind wing with shallow costal process, which is not found in any other Ephemerelloidea, and also not in any other mayfly taxa except of Baetiscidae, which are neither closely related to Ephemerelloidea, Caenoidea, Leptophlebioidea nor Ephemeroidea.

Within Vietnamellidae, the shape of male genitalia is a reliable generic character. Here, *Burmella* significantly differs from *Vietnamella*, and this is also the main reason for our attribution of *B. zhouchangfai* to the former genus. The penis lobes in *Burmella* are largely separated and thus retain a plesiomorphic condition, while in *Vietnamella* the penis lobes are medially almost entirely fused. There are no traces of paracercus in the three species of *Burmella* with preserved tip of abdomen, in contrast to the plesiomorphic character in *Vietnamella* and *Austremerellidae,* in which the paracercus is well-developed (see Table 2; *Riek, 1963*; *McCafferty & Wang, 2000*; *Jacobus & McCafferty, 2006*; *Suter & Mynott, 2013*).

Further differences between *Burmella* and *Vietnamella* can be found in the wing venation: The small marginal intercalaries in the forewing of *Vietnamella* are basally connected to the main longitudinal veins. In *Burmella,* there are also some basally free small intercalaries present, which we regard as autapomorphy of this genus. In *Vietnamella* and other Ephemerelloidea, as well as in the majority of most other families of Ephemeroptera, small marginal intercalaries are usually basally attached to the longitudinal veins. *Burmella* is plesiomorphic compared to *Vietnamella* and Austremerellidae regarding the forewing pterostigma, which is composed of simple cross veins in *Burmella*, in contrast to a "cellular" structure of the pterostigma in *Vietnamella* and Austremerellidae. Further wing characters of unclear phylogenetic information concern the course of CuP, which is smoothly curved towards wing base in *Burmella*, whereas in extant Vietnamellidae and Austremerellidae, CuP is sharply curved inwards. Also, the hind wing venation of *Burmella* is distinctly simplified to a few cross and longitudinal veins forming a single triad only (see Table 2), thus resembling the condition in *Austremerella picta* Riek, 1963, in contrast to *Vietnamella* with abundant venation (*Riek, 1963*). As the female imago of *Burmella inconspicua* **sp. nov.** is incompletely preserved without tip of abdomen, we cannot conclude on the female genitalia, but otherwise it correlates well with the female of *B. clypeata*.

## Phylogenetic position of Vietnamellidae within Ephemerelloidea

*McCafferty & Wang (2000)* placed the "*Austremerella* group" within Pantricorythinis. *Kluge (2004)*, excluding the Ephemerellinae and Timpanoginae lineages. Later, these two last lineages were strongly supported as monophyletic in a combined molecular and morphological analysis by *Ogden et al. (2009a)*. Even earlier, *McCafferty & Wang (1997)*

placed Oriental *Vietnamella* and Australian *Austremerella* in the subfamily Austremerellinae (McCafferty & Wang, 1997) within Teloganodidae McCafferty & Wang, 1997.

In a cladistic analysis of Ephemerelloidea proposed by *Jacobus & McCafferty (2006)*, Vietnamellidae and Austremerellidae are clustered as basal clades to the remaining Ephemerelloidea. Most basally, the fossil Caenozoic monotypic genus *Philolimnias sinica* Hong, 1979 (see *Hong, 1979*) was included in this cladistic analysis as sister group to all the other Ephemerelloidea (including Vietnamellidae and Austremerellidae). *Hong (1979)* and *Jacobus & McCafferty (2006)* attributed *Philolimnias* to Ephemerellidae, while *Kluge (2004)* and *Hubbard (1987)* regarded it as Euephemeroptera *incertae sedis*. Lastly, *Staniczek, Godunko & Kluge (2018)* concluded that the placement of *Philolimnias* in Ephemerelloidea is doubtful and its systematic position needs thorough re-evaluation.

Recently, *Auychinda, Sartori & Boonsoong (2020)* and *Auychinda et al. (2020)* in a molecular study of extant species confirmed the monophyly of Vietnamellidae based on well supported genetic distances between Vietnamellidae, Teloganodidae, and the genera *Dudgeodes* Sartori, 2008, *Teloganella* Ulmer, 1939, and *Teloganopsis* Ulmer, 1939. Unfortunately, representatives of Ephemerellinae were not included in these analyses.

Investigating six complete mitochondrial genomes of different species of Ephemerellinae, *Xu et al. (2020)* concluded that Ephemerellidae was the sister group to Vietnamellidae. The same result was confirmed by *Tong et al. (2022)* and *Guo et al. (2024)*. However, these both aforementioned studies should be treated with great caution regarding phylogenetic analyses, since some of the results show significant differences with most part of the phylogenetic reconstructions earlier proposed for Ephemeroptera (see above).

In summary, most molecular or combined phylogenies consider Ephemerellidae as sister group to Pantricorythini sensu *Kluge (2004)*. Also, Vietnamellidae and Austremerellidae are generally considered the most basal clades of Pantricorythini (see *McCafferty, 1991*; *Kluge, 2004*; *Ogden & Whiting, 2005*; *Ogden et al., 2009a*; *Ogden et al., 2009*; *Ogden et al., 2019*). According to these contributions, Vietnamellidae is the sister group of Austremerellidae, which is represented by the single species *Austremerella picta* (Riek, 1963), known from adults and larvae found in the Lamington National Park in Queensland, Australia (*Riek, 1963*; *Suter & Mynott, 2013*). Considering their pattern of distribution, *McCafferty & Wang (2000)* suggested Vietnamellidae and Austremerellidae as ancient Gondwanan taxa. Moreover, *Austremerella* was considered as mostly "generalized genus" in comparison to *Vietnamella*, according to *McCafferty & Wang (2000)* probably representing "the extant taxon closest to the hypothetical precursor of the entire superfamily Ephemerelloidea". Unfortunately, Vietnamellidae and Austremerellidae were not included in the otherwise most comprehensive phylogenetic study on mayflies by *Ogden et al. (2009a)*, so their position within mayflies and Ephemerelloidea awaits further clarification. This in turn, together with the accumulation of data on fossil representatives of the superfamily, may lead to refinement of biogeographic scenarios.

## Palaeoecological considerations

The larvae of extant Vietnamellidae inhabit small and middle-sized rivers and streams of moderate to fast flowing current, where they can be found on cobbles and pebbles,

sometimes also on sandy substrate. Here, they co-occur with nymphs of other mayflies, particularly other Ephemerelloidea (*Sinha et al., 2018*; *Auychinda, Sartori & Boonsoong, 2020*; *Auychinda et al., 2020*). The biology of adults is still poorly known.

The paleohabitats of the Burmese mayfly fauna most probably represented humid tropical forests (*Poinar Jr, Lambert & Wu, 2007*; *Xing et al., 2019*; *Bolotov et al., 2021*; *Yu, Neubauer & Jochum, 2021*) with dense nets of waterflows. The rich mayfly fauna documented for Burmese amber mostly represents extant families with numerous rheophilous taxa, namely Heptageniidae (*Yang et al., 2023*), Baetidae (*Sinitshenkova, 2000*; *Poinar Jr, 2011*), Prosopistomatidae (*Lin et al., 2018a*), and Leptophlebiidae (*Chen & Zheng, 2022*; *Chen & Zheng, 2024*). Two other mayfly taxa reported from this paleoecosystem, namely Australiphemeridae (*McCafferty, 1991*) and Hexagenitidae Lameere, 1917 (*Lin et al., 2018b*; *Zheng & Chen, 2023*), most probably also had rheophilous larvae, possibly distributed in lower (deltaic) sections of the waterflows with hyporhithral to epipotamal conditions. Similarly, presumably lotic or deltaic ecological niches are assumed for taxa of these families recorded from Jurassic Limestone of Solnhofen, Germany (see *Bechly, 2015*) and from the Lower Cretaceous Crato Formation in Brazil (*McCafferty, 1990*; *Storari et al., 2021*; *Storari et al., 2022*).

In addition to mayflies, the Kachin amber is rich in other groups of freshwater insects (*Rasnitsyn et al., 2016*; *Staniczek, 2019*; *Jarzembowski, Coram & Song, 2023*). Based on these data and following *Bolotov et al. (2021)*, we assume that the Kachin amber-producing forest extended from the deltaic (estuarine) areas of tropical rivers and streams to their upper sections, and also covered different types of freshwater and estuarine habitats (lakes, estuaries, coastal rivers and their river deltas). The taxonomic composition of the mayflies from Burmese amber rather indicates that their fossilisation took place along the middle and lower reaches of the waterflows, which does not exclude the presence of strong rheophilic taxa typical for the upper epirhitral portions of the streams and rivers. In that case, swarming of adults of rheophilous taxa of Ephemeroptera in lower river sections was compensated by upstream flight of fertilised females. Such upstream flight compensation (compensatory flight of lotic aquatic insects) is well described and discussed for several mayfly groups (see *Hubbard, 1991*; *Jacobus, Macadam & Sartori, 2019*; *Domínguez et al., 2023*). The rheophilic character of the Burmese fossil mayfly fauna is also well supported by the co-occurrence and dominance of the stonefly family Perlidae (*Sroka, Staniczek & Kondratieff, 2018*; *Chen, 2018*; *Jouault et al., 2022*), whose recent larvae also prefer rheophilic conditions.

## Palaeobiogeographical considerations

The current distribution of Vietnamellidae is restricted to Eastern India, Thailand, Central and Southern China. In Thailand, most species have been recorded close to the border with Myanmar (*Auychinda, Sartori & Boonsoong, 2020*; *Auychinda et al., 2020*). All fossil Vietnamellidae have been found in mid-Cretaceous Burmese amber only, among them are also species that are still undescribed (Figs. 9A–9C). Thus, both fossil and recent taxa of this family are known exclusively from Southeast Asia.

The extant worldwide distribution of Ephemerelloidea is in sharp contrast to its almost complete absence in the fossil record, and only a few extinct taxa including *Burmella*

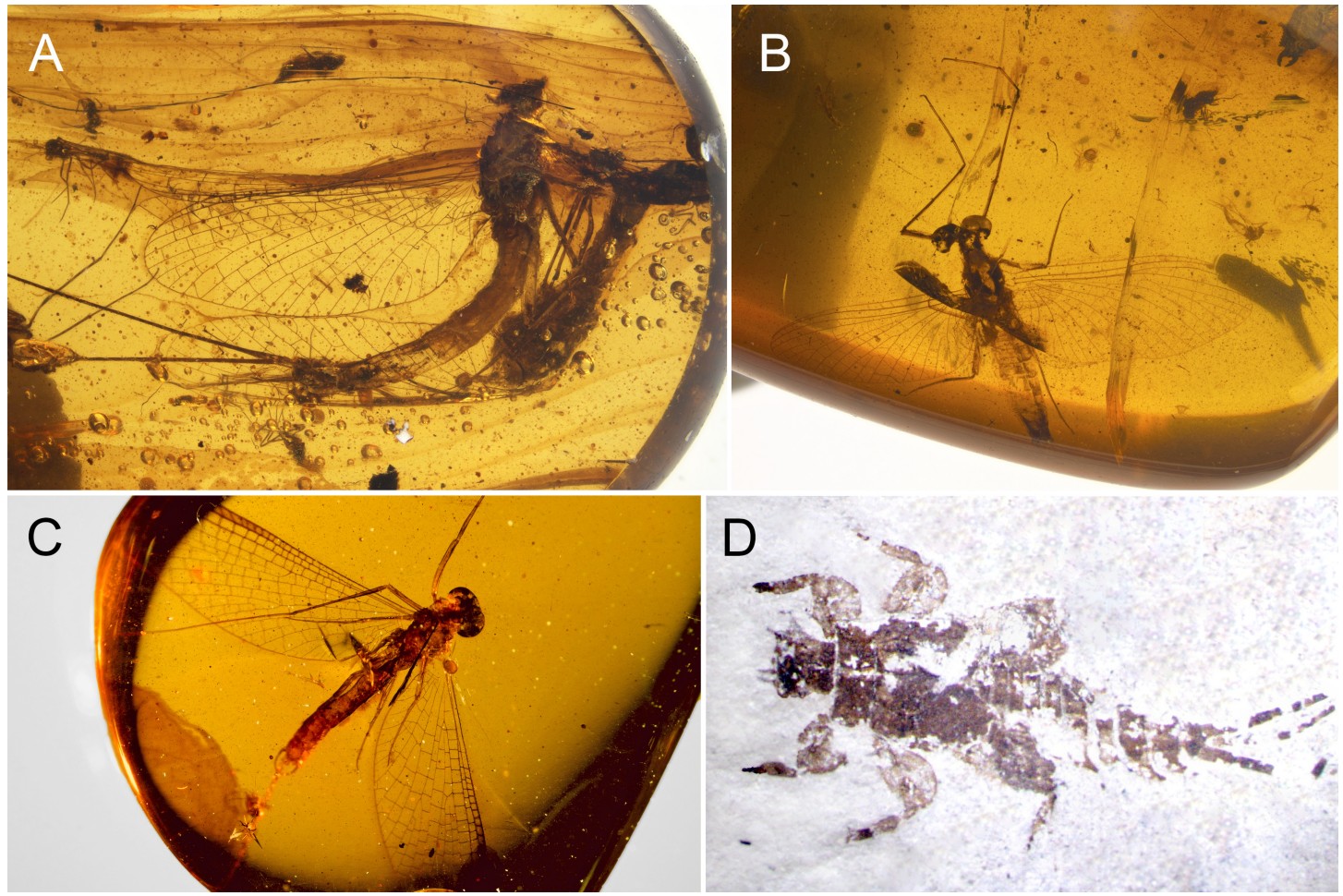

**Figure 9** **Diversity of fossil Vietnamellidae from mid-Cretaceous Burmese amber (A–C) and fossil Teloganellidae McCafferty & Wang, 2000 from the late Paleocene–early Eocene, Palana Formation, India (D).** (A–B) undescribed male imago, Patrick Müller collection (Käshofen, Germany) (A) in lateral view, (B) in dorsal view, (C) undescribed male imago in ventral view, Zhendong Lian collection (Taiwan), (D) *Teloganella gurhaensis* Agnihotri et al., 2020, larva, holotype, BSIP no. 41813, collection of the Birbal Sahni Institute of Palaeosciences (India). Without scales.

can be assigned to this group without doubt. For instance, *Staniczek, Godunko & Kluge (2018)* reinvestigated the holotype of *Ephemerella viscata* Demoulin, 1968 from Eocene Baltic amber, and transferred it to the genus *Eurylophella* Tiensuu, 1935 (Ephemerellidae: Timpanoginae Allen, 1984). In the same publication, the first confirmed record of the genus *Ephemerella* Walsh, 1862 s.l. (Ephemerellinae Lestage, 1917) was reported based on adults of both sexes from Baltic amber. In contrast to the sparse fossil record of Ephemerellidae, the recent fauna of this family includes approximately 150 species in 22 genera in the Holarctic and Oriental regions, with clear dominance in the Palaearctic and Nearctic (*Barber-James et al., 2008*; *Sartori & Brittain, 2015*).

The fossil record of Ephemerellidae and the restricted distribution of extant species in the Northern Hemisphere well reflects their Laurasian origin. According to *McCafferty & Wang (2000)*, the first major branching within the Ephemerelloidea resulted in a Laurasian clade

(*i.e.*, Ephemerellidae), and all the other lineages, which constitute a Gondwanan clade (*i.e.*, Pantricorythini sensu *Kluge, 2004*), with only recent invasions into the Holarctic, which are for instance well documented for the Nearctic Leptohyphidae (*McCafferty, 1998*; *Barber-James et al., 2008*; *Sartori & Brittain, 2015*).

However, in contrast to the scenario of *McCafferty & Wang (2000)*, suggesting the Palaeozoic origin of Ephemerelloidea, a different theory, initially put forward by *Pescador et al. (2009)*, seems more plausible: The separation of Ephemerellidae and Pantricorythini could have resulted from a vicariant event in the Jurassic at the breakup of Pangea, where the ancestors of these two lineages most likely existed even before the main continental breakout, dispersed, and initially diversified on several modern continents no later than the Early Jurassic (200–175 My). This may indicate an Early Mesozoic age for Ephemerelloidea but seems in no way older than the P–Tr extinction event. A similar scenario was also proposed for another mayfly family with Pangean origin, namely Baetiscidae (*Godunko & Sroka, 2024*).

A high probability of a Pangaean origin can also be assumed for the extinct mayfly families Hexagenitidae and Australiphemeridae, as well as for Baetidae, Leptophlebiidae and possibly Heptageniidae, which are also recorded from Burmese amber. On the other hand, among mayflies and other freshwater invertebrates in Burmese amber, there are also groups that are nowadays typical of the Southern Hemisphere (*Sroka, Staniczek & Kondratieff, 2018*; *Staniczek, 2019*; *Yu, Neubauer & Jochum, 2021*; *Bolotov et al., 2022*). Without any doubt, the ephemerelloid families Vietnamellidae and Austremerellidae also belong to these groups with possible Gondwanan origin.

The primordial supercontinent of Gondwana in the Southern Hemisphere included landmasses of South America, Africa, India, Madagascar, Antarctica, Australia, New Zealand, and New Guinea. Up to the Middle Jurassic (about 170–150 My), the active exchange of both terrestrial and aquatic biota between these landmasses was undoubtedly possible. With regards to eastern Gondwana, a biotic transfer within the Indian Plate, Burma Terrane, Greater India, Sri Lanka and Madagascar on the one hand and Australia (plus Antarctica) on the other hand, with high probability was still possible in the Early Cretaceous, but evidently not later than 135 My ago (*Wan, 2011*; *Wan, 2020*; *Wan & Zhu, 2011*; *Bolotov et al., 2022*).

The early separation of Vietnamellidae and Austremerellidae from the remaining Pantricorythini correlates well with the paleogeography of the East Gondwana. Considering that the breakup of East Gondwana started around 133–130 My, when the assembly of circum-Indian blocks and continents moved northwest away from Australia–Antarctica (*Acharyya, 1998*; *Acharyya, 2000*; *Williams et al., 2013*; *Yoshida & Hamano, 2015*; *Wan, 2020*), the ancestors of these families should have arisen even earlier, possibly during the Middle-Late Jurassic interval, with Austremerellidae being perhaps the easternmost group, while the ancient Vietnamellidae were very early separated and isolated by the drift of the Burma Terrane.

In this way, the ancestral Austremerellidae became isolated on the Australian continent. Fossil finds of mayflies in Australia are scarce. Only a few taxa have been described from the Early Cretaceous Koonwarra Fossil Bed in Victoria and the Pliocene Vegetable Creek

tin-mining field in New South Wales (*Etheridge & Olliff, 1890*; *Riek, 1954*; *Jell & Duncan, 1986*; *Jell, 2004*; *Poropat et al., 2018*). Furthermore, there are still undescribed fossil mayflies from the recently discovered Middle Miocene McGraths Flat Konservat–Lagerstätte in New South Wales (*McCurry et al., 2022*; *Baranov et al., 2024*; *Godunko & Sroka, 2024*). However, within all the listed fossils, taxa of Ephemerelloidea are neither recorded nor described. On the other hand, Cretaceous mayfly specimens from Koonwarra were mostly described as Siphlonuridae (*e.g.*, the monotypic genus *Australurus plexus* Jell & Duncan, 1986), but may have rather affinities within Leptophleboidea or even Ephemerelloidea (see also *Godunko & Sroka, 2024*). The recent Australian fauna of Ephemerelloidea is restricted to Austremerellidae as earlier noted (see *Chessman & Boulton, 1999*; *Dean & Suter, 1996*; *Suter & Mynott, 2013*; *Sartori & Brittain, 2015*). The only other finding of Ephemerelloidea within the Australasian Realm refers to *Dudgeodes celebensis* Sartori, 2008 from Sulawesi (*Sartori, Peters & Hubbard, 2008*).

Little is known about the drift of smaller elements of the Indian plate after the breakup of Gondwana, and there is no consensus on the spatial and temporal dynamics of this process. Among these smaller plate elements detached from Gondwana and migrating northwards, the Burma Terrane (also known as the Indo-Burma-Andaman block, Myanmar Microplate, Burmese Platelet, West Myanmar Terrane or West Burmese Plate) is particularly interesting in regard of the mid-Cretaceous Burmese fauna and its impact on the evolution of recent Oriental biota including freshwater insects.

It is still disputable at what time the Burma Terrane separated from East Gondwana (*Staniczek, 2019*). *Poinar Jr (2019)* discussed two hypotheses of its origin. It is either considered to have split from Australia and successively shifted to its current location in southeast Asia (*Metcalfe, 1996*; *Metcalfe, 2017*; *Hall, 2012*; *Seton et al., 2012*), or it became attached to the east of the Great Indian Plate after the break-up of Gondwana (see below).

According to the latter hypothesis, the initial separation of the Burma Terrane, together with the Indian Subcontinent and surrounding blocks from Gondwana, took place less than 135 My ago, with a subsequent final separation of the Burma Terrane from the Indian Plate, probably by a partial submergence of Greater India at around 75 My (*Bolotov et al., 2022*). Other authors have suggested that the West Burma block separated from Northern Gondwana already at the end of the Jurassic period about 160 My ago together with the Indian Subcontinent, shifting northwards through the Tethys Sea and reaching Asia around 100–90 My ago, or even later in the Santonian or Campanian (86.3–83.6 My ago) (*Heine, Müller & Gaina, 2004*; *Khin, Zaw & Aung, 2017*; *Bolotov et al., 2022*). On the other hand, *Westerweel et al. (2020)* assumed that, approximately 95 My ago, the Burma Terrane was a part of the Trans-Tethyan island arc located on a near-equatorial southern latitude, "suggesting island endemism for the Burmese amber biota". A significant clockwise rotation of the Burma Terrane would have occurred between 80–50 My, before the collision of the Late Eocene Burma Terrane–Sibumasu Terrane in Asia around 45–40 My ago (*Licht et al., 2020*; *Bandopadhyay et al., 2022*; *Bolotov et al., 2022*; *Min et al., 2022*). Finally, there is also a hypothesis that assumes a separation of the Western Burma block as early as in the Devonian (418–361 My ago) (*Metcalfe, 2017*), merging with the Asian plate already in Early Cretaceous (app. 120 My) (*Hall, Cottam & Wilson, 2011*; *Wan, 2020*).

The scenario proposed by *Bolotov et al. (2022)*, which is linked to the time-calibrated phylogeny and to statistical biogeographic models for freshwater mussels of the subfamily Parreysiinae, may also apply to other groups of freshwater insects, including mayflies. Not only does the early separation of the Vietnamellidae and Austremerellidae from the rest of Pantricorythini correspond well with this scenario, but also the diverse fossil fauna of Vietnamellidae found in the mid-Cretaceous Burmese amber, as well as the presence of extant species in the Oriental region. From there, the family would have expanded northwards to China and India in post-Eocene times. Moreover, this scenario corresponds well with the assumption of a Gondwanan origin of other groups of Pantricorythini, which now also has palaeontological evidence based on the report of the first fossil finding of the family Teloganodidae from India (*Agnihotri et al., 2020*) (Fig. 9D). *Teloganella gurhaensis* Agnihotri et al., 2020 was described as a compressed single larva from the Palana Formation (Gurha lignite mine, Rajasthan, India). Judging by its morphological characters, the placement within Teloganodidae is indeed quite possible. The geological age of the Palana Formation is indicated as Late Paleocene-Early Eocene by *Agnihotri et al. (2020)*, while *Farooq et al. (2019)* and *Mathews et al. (2020)* rather indicated an Early Palaeocene age (app. 66–56 My). At that time, the Indian Plate and Greater India were still drifting towards Asia, which allows us to consider this fossil taxon as a part of the more ancient East Gondwanan fauna.

The model proposed by *Metcalfe (1996)*, according to which the West Burma Block was separated from Australia in the Late Jurassic and docked with SE Asia in the Early Cretaceous (see also *Hall, 2012*; *Poinar Jr, 2019*; *Jouault et al., 2022*), correlates well with the initial evolution of Pantricorythini, implying an early separation and isolation of the ancestors of the basal lineages represented by Vietnamellidae and Austremerellidae.

Also, the recent and fossil mayfly fauna of Australia has noticeably less similarities with the palaeofauna preserved in Burmese amber and is also not closely related to the extant and fossil fauna of Palearctic and Oriental Asia. Therefore, models of the tectonic evolution of the East Gondwana postulating the initial separation of Australia and Antarctica (isolation of the family Austremerellidae) and their subsequent detachment from the common landmass comprising the Indian Subcontinent and Burma Terrane (the ''biotic ferry'' for Vietnamellidae and part of Teloganodidae), correlate well with the evolution and current biogeography of these basal groups of Pantricorythini.

There is however no doubt that the colonisation of the Burma Terrane by groups of Laurasian origin was only possible at a time, when this block was approaching the Asian part of Laurasia and thus, the influence of Northern faunal elements increased during the Late Cretaceous. New groups of aquatic insects then probably migrated southwards to the Burma Terrane by passive transfer of winged stages with the winds or even by drift. However, due to the calibration of Burmese amber to 99 My based on UePb zircon dating, there is no doubt at all that the Burma Terrane already at that time must have already been reachable for Laurasian elements. As the time frame for the tectonic shift suggested by *Bolotov et al. (2022)* implies a location of the Burma Terrane far south of the equator 100 My ago, either the calculated time of the continental shift would be wrong or

those groups to be considered of Laurasian origin already at that time had migrated south towards Gondwana or in fact were of Pangean distribution.

This dispersal scenario may not only apply for recent families with Laurasian origins (*e.g.*, Prosopistomatidae, Heptageniidae, Baetidae: Palaeocloeoninae), but also for fossil Pangaean groups (Australiphemeridae and Hexagenitidae), nevertheless it does not explain the spatial and temporal collisions described here for basal Pantricorythini. The alternative would be a wider distribution of Pantricorythini in the northern hemisphere before the Cretaceous, which is not yet supported by the fossil evidence of the Ephemerelloidea analysed in detail above. As for other insect groups (including aquatic and terrestrial), early faunal elements are supposed to have Gondwanan origin with later influence of Laurasian taxa as suggested by *Poinar Jr (2019)*, *Staniczek (2019)*, *Jouault et al. (2022)* and *Wichard (2023)*.

The discovery of new fossil sites from the Oriental region and the Indian subcontinent may fill the gaps in our present knowledge and may help in a more accurate reconstruction of spatial and temporal dispersal of biota during the Cretaceous. For example, *Zheng et al. (2018)* reported on a unique amber biota from the Upper Cretaceous of Tilin in central Myanmar, dated to the uppermost Campanian (app. 72.1 My). Likewise, *Xing & Qiu (2020)* pointed to a little known but diverse arthropod fauna from the mid-Cretaceous Hkamti amber in northern Myanmar with an approximate age of 110 My. In both fossil sites, undescribed adults of aquatic insect groups (*e.g.*, Chironomidae and Ceratopogonidae, and some Trichoptera) were listed, so the discovery of mayflies in these sites may seem well possible.

## CONCLUSIONS

Originally established as a monotypic extant mayfly family, the Oriental Vietnamellidae now comprise also fossil species. The diversity of the genus *Burmella*, initially described based on two species from mid-Cretaceous Burmese Amber, has been much larger than previously thought, and in the current study we are adding two more new species. *Burmella inconspicua* **sp. nov.** is established based on a female imago. It differs well from the previously described *B. paucivenosa* in lacking an anterolaterally expanded clypeus. *B. zhouchangfai*, originally described as type species of the genus *Burmaheptagenia* in the family Heptageniidae, is transferred to *Burmella* within Vietnamellidae. Therefore, a new synonymy *Burmella* = *Burmaheptagenia* **syn. nov.** is established. The increase in the number of fossil representatives of Vietnamellidae has led us to redefine and clarify the generic diagnosis of *Burmella*, and to analyse in detail the morphological differences to extant species of the genus *Vietnamella*. Cretaceous Vietnamellidae differ well from recent representatives by the venation of the forewings, shape and venation of hind wings, and the structure of male genitalia.

Well-preserved adult specimens of Vietnamellidae in mid-Cretaceous Burmese Amber also provided the opportunity to discuss the evolution of the basal lineages of Ephemerelloidea (and Pantricorythini in particular), as well as the different geological scenarios on the Gondwanan influence in shaping the extant fauna of the Oriental region

with emphasis on the island character of the Burma Terrane in the Cretaceous and its role as a "biotic ferry" (together with the Indian Subcontinent) for taxa travelling from Eastern Gondwana to Asia. Based on current insights into the systematics and phylogeny of Ephemerelloidea, we consider several scenarios for history of the fossil fauna in Burmese Amber and in the extant fauna of the Oriental region. Our conclusions are reconciled with current studies in palaeontology, geology and tectonics, which show the Gondwanan origin of the Burmese terrane and its movement northwards across the Tethys Ocean in the Cretaceous, together with the older fauna of the East Gondwana, and of its merging with Laurasian biotic elements approaching to Asia later in the Paleogene.

## ACKNOWLEDGEMENTS

We are grateful to Patrick Müller, Käshofen, Germany, for the donation of the type specimen of *Burmella inconspicua* **sp. nov.** and other fossil Vietnamellidae from mid-Cretaceous Burmese amber. We also thank Zhendong Lian (Tainan City, Taiwan) and Priya Agnihotri (Birbal Sahni Institute of Palaeosciences, India) for the photographs of fossil Ephemerelloidea. We are grateful to Boonsatien Boonsoong (Kasetsart University, Bangkok, Thailand) for access to original photographs of extant adults of Vietnamellidae. We would also like to thank Kateřina Bláhová for the help with line drawings and Milan Pallmann for the help with macro photography. Comments from anonymous reviewer helped to improve the manuscript.

### Funding

Open access funding was provided by the University of Lodz (Poland). The stay of Roman J. Godunko at SMNS has been made possible through a fellowship granted by the Alexander von Humboldt Foundation in 2023–2024. This study was also realized with institutional support of the Institute of Entomology (Biology Centre of the Czech Academy of Sciences) RVO: 60077344 for Roman J. Godunko. Comparative investigation of the wing morphology of Mesozoic mayfly taxa was supported by the Grant Agency of the Czech Republic (No. 24-11498S) for Roman J. Godunko. The funders had no role in study design, data collection and analysis, decision to publish, or preparation of the manuscript.

### Grant Disclosures

The following grant information was disclosed by the authors:
The University of Lodz (Poland).
Alexander von Humboldt Foundation in 2023–2024.
The Institute of Entomology (Biology Centre of the Czech Academy of Sciences) RVO: 60077344.
The Grant Agency of the Czech Republic: No. 24-11498S.

### Competing Interests

The authors declare there are no competing interests.

## Author Contributions

- Roman J. Godunko conceived and designed the study, analyzed the data, prepared figures and/or tables, authored or reviewed drafts of the article, prepared the first draft of the manuscript, and approved the final draft.
- Nadhira Benhadji analyzed the data, prepared figures and/or tables, authored or reviewed drafts of the article, and approved the final draft.
- Alexander Martynov analyzed the data, authored or reviewed drafts of the article, and approved the final draft.
- Zhi-Teng Chen analyzed the data, authored or reviewed drafts of the article, and approved the final draft.
- Xuhongyi Zheng analyzed the data, authored or reviewed drafts of the article, and approved the final draft.
- Arnold H. Staniczek conceived and designed the study, analyzed the data, prepared figures and/or tables, authored or reviewed drafts of the article, and approved the final draft.

## Data Availability

Three holotypes mentioned in this publication are stored in the SMNS amber collection (Stuttgart, Germany), including the holotype of *Burmella inconspicua* **sp. nov.** under the inventory number SMNS BU-385.

The holotype of *B. zhouchangfai* is deposited at the Insect Collection of Jiangsu University of Science and Technology (Jiangsu, China), under the inventory number CZT-EPH-MA5.

The authors declare that to their knowledge, the fossil material reported in this study was not involved in armed conflict and ethnic strife in Myanmar. The fossil specimen described here as a new species was collected and acquired in Myanmar in full compliance with the laws of this country before 2017.

## New Species Registration

The following information was supplied regarding the registration of a newly described species:

Publication LSID: urn:lsid:zoobank.org:pub:B1FAF587-9EB2-4CDB-BD71-5AE0C9C6AD78.

*Burmella inconspicua* **sp. nov.** LSID: urn:lsid:zoobank.org:act:6B45EB0E-8DF7-49D8-B317-E564009C00DC.

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
