# Peer review of "A new species and new generic synonymy in the family Vietnamellidae (Insecta: Ephemeroptera) from mid-Cretaceous Burmese amber with notes on ancient dispersal across East Gondwana"

_PeerJ, doi:10.7717/peerj.19048_

## Round 0.1 · original submission · Minor Revisions

Please, take all the comments into account and address them.

Reviewer 1 ·

Basic reporting

The article is well developed in all the areas. The English is clear and unambiguous. Literature references are sufficient. The article is also well structured.

Experimental design

The investigation was rigorously performed. The methods were sufficient and detailed. The question was relevant and meaningful.

Validity of the findings

The article describes a new species of a fossil mayfly of scientific interest. The data use for the description was provided. Finally, the conclusions are in accordance with the entire writting.

Additional comments

After reviewing the paper untitled “A new species and new generic synonymy in the family Vietnamellidae (Insecta: Ephemeroptera) from mid-Cretaceous Burmese amber with notes on ancient dispersal across East Gondwana” I consider that it can be published.
The paper is well written in general. The authors also provide a solid introduction and the materials and methods are well developed. The authors show good photographs and drawings that support the information they give. However, I detected a few flaws that I mention in the attached Word file.
The comments are diverse, which involve simple spelling or syntax changes to slightly more profound ones, such as the description of the new taxon and the possibility of using the so called paratxonomy (as formal taxon, for example), that is in function of the probability of knowing if the previous described species based on males are or not, part of the same species.

Annotated reviews are not available for download in order to protect the identity of reviewers who chose to remain anonymous.

Reviewer 2 ·

Basic reporting

Clear and well-organized

Experimental design

Well designed

Validity of the findings

no comment

Additional comments

The manuscript is well-written and organized, with high-quality figures. It provides a comprehensive of the the family Vietnamellidae from mid-Cretaceous Burmese amber and deserves to be published in PeerJ.

Reviewer 3 ·

Basic reporting

Clear, unambiguous, professional English was used throughout the text. Minor comments to improve the readability:
Improve the transition between the paragraphs in the introduction, specifically between lines 91-92.
Search for all the occurrences of the word "recent" and double-check their case setting. They are typed starting in capital letters.

Line 61: Keywords are generally words that improve the searchability of the research but are different from the words in the title. Replace Burmese amber, Cretaceous, and new species since they appear on the title already. Alphabetically arranged the terms here.

The introduction and background sufficiently gave context to the study. Sufficient background information on Vietnamella species were provided. Minor comments that should be addressed as a taxonomic paper:
Kindly add in the bibliography all the references for the taxon authorities (e.g., Tong 2020, Hsu 1936, Tiensuu, 1935). If possible, also add the taxon authorities on the subphylum to the family level. In the context of the discussion where order-family identification and relationships are discussed, it is important to cite these papers.

The figures were relevant, high quality, and well labeled/described in the text. Minor comment on the consistency of reporting:
-Fig 1A, B and Fig. 2A-B formats are used simultaneously. Choose one and stick with it. Adjust for the other figure mentions.

Experimental design

The last paragraph of the introduction provided a good context and outline of all the research questions and results of the manuscript.

Validity of the findings

I am not sure if the lines 155-166 are necessary since this [PeerJ] is not a new journal. By custom and for clarity, the link of the Zoobank registration is usually placed immediately on the subheading of the taxonomic act to identify each act and valid registry: e.g., the new specieZoobanknk link should be on the new species section. This is especially useful in this paper, where several taxonomic acts are done (new species, new generic synonymity).

Conclusions are well stated. The discussion of the results is comprehensive.

---

## Round 0.2 · Minor Revisions

Thank you very much for addressing all reviewers' comments. The manuscript is almost ready for publication. I would only request that the statement regarding the Burmese amber be added. According to the journal requirements, if the amber piece was obtained before 2017, an ethical statement along the following lines should be mentioned: "The fossil was collected in full compliance with the laws of Myanmar and XX in YEAR. To avoid any confusion and misunderstanding, all authors declare that to their knowledge, the fossil reported in this study was not involved in armed conflict and ethnic strife in Myanmar, and was acquired prior to 2017".

Reviewer 1 ·

Basic reporting

The article uses well-written English; the literature references are suffiicient for the context and background; the article provides professional tables and figures.

Experimental design

The article is based on original primary research aims; the research question is well defined and relevant; rigourous investigation was permformed to high technical standards; methods are described in detail

Validity of the findings

Impact and novelty is assessed; all underlying data were provided; conclusions are ad hoc with supporting results

Additional comments

The authors have improved the article in all previous comments made in the first round. I consider that the article meets the standards that the journal requests to be published.

Reviewer 3 ·

Basic reporting

The authors addressed the previous comments. No further recommendations here.

Experimental design

The authors addressed the previous comments. No further recommendations here.

Validity of the findings

The authors addressed the previous comments. No further recommendations here.

Additional comments

The authors addressed the previous comments. No further recommendations here.

---

## Round 0.3 · accepted · Accept

Thank you for addressing all the comments. I think the manuscript is ready for publication.